# A description of the first open source community release of MISTRA-v9.0: a 0D/1D atmospheric boundary layer chemistry model

Josué Bock[1,2], Jan Kaiser[1], Max Thomas[1,3], Andreas Bott[4], and Roland von Glasow[†]

[1]Centre for Ocean and Atmospheric Sciences, School of Environmental Sciences, University of East Anglia, NR4 7TJ, Norwich, UK
[2]Now at EDYTEM, Université Savoie Mont-Blanc, CNRS, 73000 Chambéry, France
[3]Now at the Department of Physics, University of Otago, P.O. Box 56, Dunedin 9054, New Zealand
[4]Institute of Geosciences, University of Bonn, Bonn, Germany
[†]deceased, 6 September 2015

**Correspondence:** Josué Bock (josue.bock@univ-smb.fr)

**Abstract.** We present MISTRA-v9.0, a one dimensional (1D) and box (0D) atmospheric chemistry model. The model includes a detailed particle description with regards to the microphysics, gas-particle interactions, and liquid phase chemistry within particles. Version 9.0 is the first release of MISTRA as an open-source community model. A major review of the code has been performed along with this public version release to improve the user-friendliness and platform-independence of the model. The purpose of this public release is to maximise the benefit of MISTRA to the community by making the model freely available and easier to use and develop. This paper presents a thorough description of the model characteristics and components. We show some examples of simulations reproducing previous studies with MISTRA, finding that version 9.0 is consistent with previous versions.

## 1 Introduction

### 1.1 Scientific context and purpose of the model

Atmospheric aerosols are a major component of the Earth climate system. They significantly affect the radiative balance of the atmosphere, through direct (scattering and absorption) and indirect effects (cloud properties modification) (Carslaw et al., 2010; Boucher et al., 2014; Bellouin et al., 2020). However their concentrations, chemical and physical properties are still insufficiently constrained, and the variability associated with their effects is dominant in the uncertainties of climate projections (Bender, 2020). Atmospheric particles also have a fundamental role in the chemistry of the atmosphere, since they offer a large surface area and volume for (photo)-chemical reactions to occur (Andreae and Crutzen, 1997; Finlayson-Pitts, 2009; George et al., 2015; Simpson et al., 2015; Seinfeld and Pandis, 2016; Kanakidou et al., 2018). Other impacts include the reduction of visibility (see for instance Seinfeld and Pandis (2016, Chap. 15); Zhang et al. (2020) and ref. therein) and health effects of pollution (e.g. Pöschl (2005), Molina et al. (2020) and ref. therein).

Numerical models are essential tools to help understand the relevant processes, and make projections of their evolution in a changing climate (Ervens, 2015). While global three-dimensional (3D) models, and specifically Earth System Models (ESMs) are well suited for climate simulation, the high computing cost of coupled physical-microphysical-chemical processes modelling limits the space resolution of such models. Currently, a kilometer-scale resolution is already very challenging, thus preventing fully resolved approaches for subgrid scale processes, such as turbulence. Conversely, limited-area and one-dimensional (1D) models can reach sufficiently fine resolution for process-resolving simulations. Ultimately, box (0D) models are designed to focus only on a single grid cell processes, further reducing the computing cost as compared to 1D models. In turn, the results obtained with such models can be used to develop parameterisations for use in 3D models.

Whatever the model, a crucial step is the validation based on field measurements. Balloon or flight surveys provided valuable data for this purpose, but the number of investigated parameters is necessarily limited, with many uncertainties and unknown values. Another useful tool for atmospheric chemistry and physics understanding are the atmospheric simulation chambers (see for instance https://www.eurochamp.org, last accessed 01-July-2021). These platforms enable the simultaneous measurement of a large number of chemical species and associated physical characteristics, in a constrained volume. The resulting datasets are highly valuable to validate models. In turn, numerical models are complementary tools to help understand and interpret measured results.

In this paper, we present the 1D boundary layer chemistry model MISTRA-v9.0, including size-resolved aerosol processes as well as particle-chemistry interaction. MISTRA-v9.0 also includes a box-model (0D) configuration, which can be adapted for atmospheric simulation chamber applications. In Sect. 1.2 and 1.3, we give a brief history of the MISTRA model, then an overview on the recent developments presented in this paper. Section 2 gives a thorough description of processes implemented in the model. Section 3 presents practical and technical aspects of MISTRA-v9.0 with the main settings, while a set of example simulations reproducing previous studies settings and configurations is presented in Section 4, to show the consistency of MISTRA-v9.0 with previous results.

## 1.2 History of MISTRA and reference publications

The MISTRA model was originally designed to study the MIcrophysics in STRAtus clouds, and was written based on a fog model (MIFOG: Bott et al. (1990); Bott and Carmichael (1993); von Glasow and Bott (1999)). Bott et al. (1996) developed the first version of MISTRA, for the simulation of cloud microphysics in the marine boundary layer (MBL). The unique feature of this model is the use of a two-dimensional particle distribution, with one dimension accounting for dry particle radius, and the second dimension for the total particle radius. Based on this first version of MISTRA, Bott (1997) further included typical particle distributions of urban and rural aerosols for the study of MBLs influenced by continental air masses, and assessed the radiative forcing of stratiform clouds. The radiation code used in MISTRA, called PIFM1 (Practical Improved Flux Method, developed by Zdunkowski et al., 1982), was updated by Loughlin et al. (1997) and the new radiation code, PIFM2, was evaluated. The collision-coalescence process was implemented in MISTRA by Bott (2000, 2001). Bott (1999a) adapted the chemistry module from Bott and Carmichael (1993) in MISTRA, with special emphasis on sulfur chemistry, and studied the retroaction of cloud processing over the microphysics in Bott (1999b). Meanwhile, von Glasow (2000) developed

another chemistry module for MISTRA, with special emphasis on halogen chemistry, and presented the results for cloud free
(von Glasow et al., 2002a) and cloudy MBLs (von Glasow et al., 2002b). Our paper develops the branch of MISTRA based on
von Glasow (2000), whose development and application until 2015 took place under the lead of Roland von Glasow.

From the early 2000s to the mid 2010s, MISTRA was regularly improved with respect to the chemistry and the gas-particle
interactions, and was used in several studies, many of them with a focus on tropospheric halogen chemistry. MISTRA was used
to investigate the influence of organic coating at the surface of sea salt particles over boundary layer chemistry, and especially
on bromine and chlorine chemistry in the aqueous phase (Smoydzin and von Glasow, 2007). A major development was the
introduction of a module for aerosol nucleation which significantly improved the iodine chemistry (Pechtl et al., 2006, 2007).
The gas-phase chemical mechanism was updated by Sommariva and von Glasow (2012).

Over the years, numerous modelling studies were performed using MISTRA (von Glasow and Crutzen, 2004; Pechtl and
von Glasow, 2007; Lawler et al., 2009; Jones et al., 2010; Joyce et al., 2014) including alternative model configurations where
the chemistry was computed in a zero-dimension (0D) atmospheric box-model mode (Buys et al., 2013), and a 0D chamber
mode (Buxmann et al., 2015). MISTRA was also adapted to model specific environments such as volcanic plumes (Aiuppa et
al., 2007; Bobrowski et al., 2007, 2015) and polar conditions (Piot and von Glasow, 2008, 2009; Buys et al., 2013). MISTRA
was also used to simulate the boundary layer chemistry over the Dead Sea after implementing a calculation of chemistry in this
specific liquid medium and an explicit calculation of sea-air gas exchanges (Smoydzin and von Glasow, 2009). A module for
firn chemistry was developed and coupled to MISTRA to specifically address the influence of chemical reactions occurring in
the snowpack on the oxidative capacity of the atmosphere in snow covered regions (Thomas et al., 2011, 2012). In this study,
we present a selection of a few specific model settings reproducing previous work, to compare the original results with those
obtained with MISTRA-v9.0.

## 1.3   Recent developments and public release

The previous (non-public) version of MISTRA (v7.4.1) included the update of the gas-phase chemical mechanism by Sommariva and von Glasow (2012). A version 8 featuring an alternative chemical bin definition was partly developed but not
completed, thus explaining the current version number. More information about past versions of MISTRA can be found
in the preface of the manual (https://github.com/Mistra-UEA/Mistra/blob/master/doc/manual_v9.0.pdf). Since 2015, significant efforts have been devoted to release MISTRA as an open-source model, including major technical improvements.
The original code, written in Fortran77, has been updated to Fortran90 to ease future maintenance and developments. To
improve robustness and portability of the code, intensive controls throughout the code have been performed to track issues, fix bugs, and conform to strict coding rules (Metcalf et al., 2004) and coding standards (see for instance http://www.
umr-cnrm.fr/gmapdoc/IMG/pdf/coding-rules.pdf and http://www.reading.ac.uk/physicsnet/units/3/3phss/F90Style.pdf, last accessed 26/10/2021). This was achieved with the help of the Fortran analyser Forcheck (v14.6, no longer distributed), as well
as standard code check options of compilers. Computing efficiency has also been improved by factorising parts of code, and
re-indexing arrays to respect column-major order in Fortran (i.e. innermost do-loops should be leftmost indexes). The chemical "Kinetic PreProcessor" (KPP: Damian et al., 2002; Sandu and Sander, 2006) has been updated to the latest version 2.2.3

(https://people.cs.vt.edu/~asandu/Software/Kpp/ last accessed 23 June 2021) with minor tuning for use in MISTRA (see the *Code availability* section at the end of the paper). Overall, several technical developments have been implemented to make the model as user-friendly as possible, and easier to adopt. The model code of MISTRA-v9.0 now has improved readability, documentation, and is available under licence EUPL-v1.1 on https://github.com/Mistra-UEA.

## 2 Scientific description

### 2.1 Overview of the model components

MISTRA is a one-dimensional model of the MBL. The vertical grid is separated into three regions: the lowest part is made of 100 layers with a constant thickness of 10 m, followed by 50 layers with logarithmically equidistant layers up to 2000 m height. The third region is a constant atmosphere whose characteristics are based on the standard atmosphere. It extends up to 50 km height and is only used for radiation calculations. These vertical grid settings (number and thickness of layers) can be easily configured if required.

Figure 1 shows schematically the most important processes that are included in the model for a cloudy MBL. The meteorological and microphysical part consists of the boundary layer model MISTRA described in detail by Bott et al. (1996) and Bott (1997). The most important processes are turbulent mixing, condensation, evaporation and radiative heating. Apart from dynamics and thermodynamics, MISTRA includes a detailed microphysical module that calculates particle growth explicitly and includes feedbacks between radiation and particles. The radiative-transfer parameterisation is a standard two-stream code using 6 spectral bands for visible and 12 bands for infrared radiation (Loughlin et al., 1997). A chemistry module computes the atmospheric chemistry in the gas phase and in the particles. Gas phase chemistry is active in all model layers; aerosol chemistry only in layers where the relative humidity has been greater than the deliquescence humidity and not dropped below the crystallisation humidity (as discussed in Sect. 2.3.3). When a cloud forms, cloud droplet chemistry is also active. Fluxes of sea salt aerosol and gases from the ocean are included (see Sect. 2.3.6). A nucleation module is also included to account for new particles nucleated from the gas phase species (see Sect. 2.3.7).

### 2.2 Meteorology, microphysics and thermodynamics

The model is one-dimensional, thus all variables are taken to be horizontally homogeneous. The set of prognostic variables comprises the horizontal components of the wind speed $u$ and $v$, the specific humidity $q$, and the potential temperature $\theta$. The Boussinesq approximation is applied and the air pressure is derived from the large scale hydrostatic equilibrium.

The set of governing equations for these prognostic variables is:

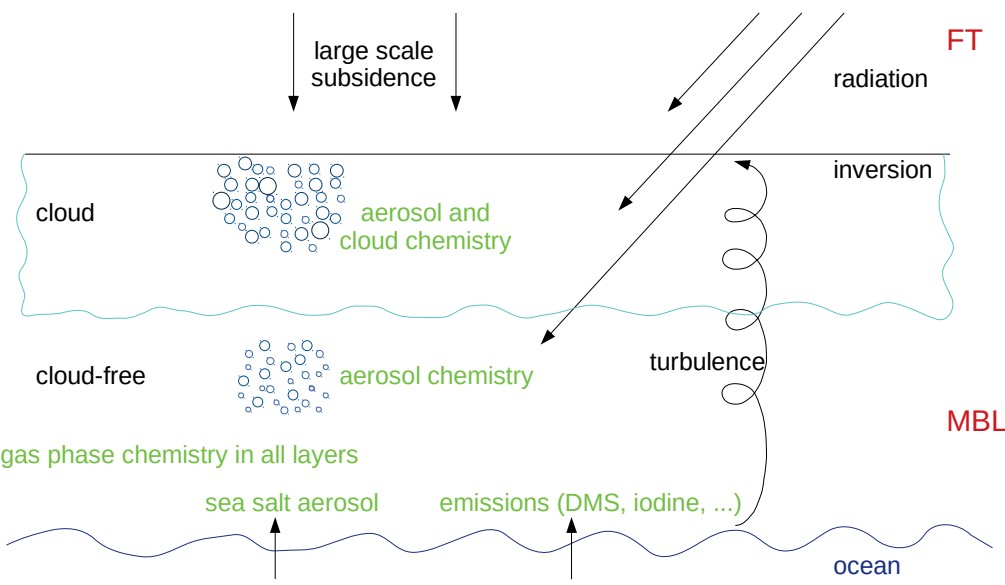

**Figure 1.** Schematic depiction of the most important processes included in the one-dimensional boundary layer model MISTRA. The free troposphere and marine boundary layer are denoted as FT and MBL, respectively.

$$115 \quad \frac{\partial u}{\partial t} = -w\frac{\partial u}{\partial z} + \frac{\partial}{\partial z}\left(K_{\mathrm{m}}\frac{\partial u}{\partial z}\right) + f_{\mathrm{c}}(v - v_{\mathrm{g}}) \tag{1}$$

$$\frac{\partial v}{\partial t} = -w\frac{\partial v}{\partial z} + \frac{\partial}{\partial z}\left(K_{\mathrm{m}}\frac{\partial v}{\partial z}\right) - f_{\mathrm{c}}(u - u_{\mathrm{g}}) \tag{2}$$

$$\frac{\partial q}{\partial t} = -w\frac{\partial q}{\partial z} + \frac{\partial}{\partial z}\left(K_{\mathrm{h}}\frac{\partial q}{\partial z}\right) + \frac{C}{\rho} \tag{3}$$

$$\frac{\partial \theta}{\partial t} = -w\frac{\partial \theta}{\partial z} + \frac{\partial}{\partial z}\left(K_{\mathrm{h}}\frac{\partial \theta}{\partial z}\right) - \left(\frac{p_0}{p}\right)^{\frac{R_{\mathrm{a}}}{c_p}}\frac{1}{c_p\rho}\left(\frac{\partial E_{\mathrm{n}}}{\partial z} + LC\right) \tag{4}$$

where $f_{\mathrm{c}}$ is the Coriolis parameter, $u_{\mathrm{g}}$ and $v_{\mathrm{g}}$ are the geostrophic wind components, $K_{\mathrm{m}}$ and $K_{\mathrm{h}}$ are the turbulent exchange

coefficients for momentum and heat, $L$ is the latent heat of condensation, $C$ the condensation rate, $\rho$ the air density, $p$ the air pressure, $p_0$ the air pressure at the surface, $R_{\mathrm{a}}$ the specific gas constant for dry air, $c_p$ the specific heat of dry air at constant pressure, and $E_{\mathrm{n}}$ the net radiative flux density, respectively. The first term on the right of each equation is the large scale subsidence. Strictly, in a one-dimensional framework, the vertical velocity $w$ should be zero everywhere, otherwise this implies a downward mass transport (for $w < 0$) without lateral outflow at the bottom of the 1D model column as would occur in the real

atmosphere. Therefore the mass balance is violated if subsidence is included. However, including subsidence is essential for modelling stratiform cloud evolution (e.g. Driedonks and Duynkerke, 1989). In runs where only aerosol chemistry is studied, i.e. in runs without clouds, the vertical velocity is set to zero ($w = 0$) in the model to avoid this problem, while for the cloud runs subsidence is usually included.

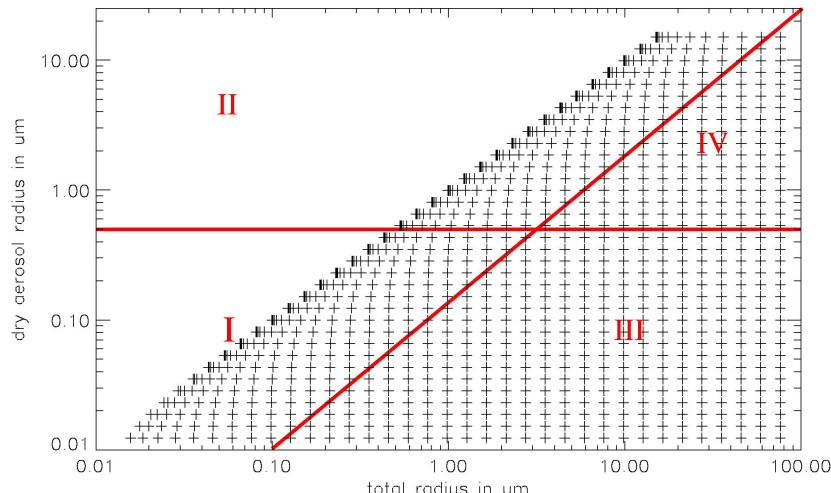

**Figure 2.** The two-dimensional particle spectrum as a function of the dry aerosol radius $a$ and the total particle radius $r$. Added are the chemical bins. I: sulfate aerosol bin, II: sea salt aerosol bin, III: sulfate cloud droplet bin, IV: sea salt droplet bin. For simplicity a 35 by 35 bin grid is plotted, in the model 70 x 70 bins are used.

Turbulence is treated with the level 2.5 model of Mellor and Yamada (1982) with the modifications described in Bott et al. (1996) and Bott (1997). The turbulent exchange coefficients $K_\mathrm{m}$ and $K_\mathrm{h}$ are calculated via stability functions $S_\mathrm{m/h}$ and $G_\mathrm{m/h}$, where the subscript m stands for shear and h for buoyancy production. The prognostic equation for the turbulence kinetic energy (TKE) $e$ is:

$$\frac{\partial e}{\partial t} = -w\frac{\partial e}{\partial z} + \frac{\partial}{\partial z}\left(K_\mathrm{e}\frac{\partial e}{\partial z}\right) + \frac{(2e)^{3/2}}{l}\left(S_\mathrm{m}G_\mathrm{m} + S_\mathrm{h}G_\mathrm{h} - \frac{1}{16.6}\right) \tag{5}$$

assuming a constant dissipation ratio (last term on the right). For more details and an explanation of the calculation of the mixing length $l$, the exchange coefficient $K_\mathrm{e}$ for the TKE, and the functions $S_\mathrm{m/h}$ and $G_\mathrm{m/h}$ see Mellor and Yamada (1982), Bott et al. (1996), and Bott (1997).

The microphysics is treated using a joint two-dimensional particle size distribution function $f(a,r)$ where $a$ is the dry aerosol radius the particles would have if no water were present in the particles, and $r$ is the total particle radius. The two-dimensional particle grid is divided into 70 logarithmically equidistant spaced dry aerosol classes. The minimum dry aerosol radius is generally set to 0.005 μm and the maximum radius 15 μm. Choosing these values allows to account for all accumulation mode particles and most of the coarse particles. The minimum and maximum, as well as the number of bins for both dimensions of the particle spectrum are adjustable. Each of the 70 dry aerosol classes is associated with 70 total particle radius classes, ranging from the actual dry aerosol radius up to 60 μm (150 μm in cloud runs). See Figure 2 for a depiction of 2D particle grid. The prognostic equation for $f(a,r)$ is:

$$145 \quad \frac{\partial f(a,r)}{\partial t} = -w \frac{\partial f(a,r)}{\partial z} + \frac{\partial}{\partial z}\left(K_{\mathrm{h}}\rho \frac{\partial f(a,r)/\rho}{\partial z}\right) - \frac{\partial}{\partial z}\left(w_{\mathrm{t}} f(a,r)\right) - \frac{\partial}{\partial r}\left(\dot{r} f(a,r)\right) \quad (6)$$

Again, subsidence is the first term on the right, followed by turbulent mixing, particle sedimentation ($w_t$ is the sedimentation velocity) and changes in $f$ due to particle growth ($\dot{r} = dr/dt$). The 2D particle spectrum is initialised with distribution depending on the type of aerosol chosen (see Bott, 1997, and ref. therein). Currently, particle distributions are provided for typical marine, rural and urban air masses. Other distributions are available for specific studies, such as a polar distribution (see for instance Buys et al., 2013, and the corresponding example simulation). Particles are initialised with a water coating according to the equilibrium radius of the dry nucleus at the ambient relative humidity. During the time integration, particle growth is calculated explicitly for each bin of the 2D particle spectrum using the growth equation after Davies (1985) (see also Bott et al., 1996):

$$r\frac{dr}{dt} = \frac{1}{C_1}\left[ C_2\left(\frac{S_\infty}{S_{\mathrm{r}}} - 1\right) - \frac{F_{\mathrm{d}}(a,r) - m_{\mathrm{w}}(a,r) c_{\mathrm{w}} dT/dt}{4\pi r}\right] \quad (7)$$

with the ambient supersaturation $S_\infty$ and the supersaturation at the droplet's surface $S_{\mathrm{r}}$ according to the Köhler equation:

$$S_{\mathrm{r}} = \exp\left[\frac{A}{r} - \frac{Ba^3}{r^3 - a^3}\right]. \quad (8)$$

where factors $A$ and $B$ account for the Kelvin effect and the solute effect, respectively.

The change in particle radius is not determined by changes in water vapour saturation alone, but also by the net radiative flux at the particle's surface $F_{\mathrm{d}}(a,r)$, that leads to temperature changes and therefore to condensation or evaporation. The constants 160 $C_1$ and $C_2$ in equation (7) are:

$$C_1 = \rho_{\mathrm{w}} L + \frac{\rho_{\mathrm{w}} C_2}{D'_{\mathrm{v}} S_{\mathrm{r}} \rho_{\mathrm{s}}} \qquad\qquad C_2 = k'T\left[\frac{L}{R_{\mathrm{v}}T} - 1\right]^{-1}, \quad (9)$$

$m_{\mathrm{w}}(a,r)$ is the liquid water mass of the particle, $c_{\mathrm{w}}$ and $\rho_{\mathrm{w}}$ are the specific heat and density of water, $\rho_{\mathrm{s}}$ is the saturation vapour density and $R_{\mathrm{v}}$ the specific gas constant for water vapour. The thermal conductivity $k'$ of moist air and the diffusivity of water vapour $D'_{\mathrm{v}}$ have been corrected for gas kinetic effects following Pruppacher and Klett (1997) (their equations 13.20 165 and 13.14, respectively). For the accommodation coefficient of water (condensation coefficient), a value of $\alpha_{\mathrm{c}} = 0.036$ is used (see table 5.4 in Pruppacher and Klett (1997) for a compilation of measured $\alpha_c$ values; in table 13.1 they use $\alpha_{\mathrm{c}} = 0.036$ as "best estimate").

The condensation rate $C$ in equation (4) is determined diagnostically from the particle growth equation (7).

Collision-coalescence processes are not included in the model because this leads to difficulties when redistributing the 170 chemical species in the particles. A version of MISTRA including collision-coalescence without considering chemistry does exist (Bott, 2000), and this limitation of MISTRA-v9.0 is discussed in Sect. 4.1.2.

For the calculation of the radiative fluxes, a $\delta$-two stream approach is used (PIFM radiative code: Zdunkowski et al., 1982; Loughlin et al., 1997). The radiative fluxes are used for calculating heating rates and the effect of radiation on particle growth. The radiation field is calculated with the aerosol/cloud particle data from the microphysical part of the model, so feedbacks between radiation and particle growth are fully implemented. The calculation of photolysis frequencies is described in Sect. 2.3.5.

## 2.3 Chemistry

The multiphase chemistry module comprises chemical reactions in the gas phase as well as in deliquescent aerosol and cloud particles. Transfer between gas and aqueous phase and surface reactions on particles are also included. The reaction set was based on that of Sander and Crutzen (1996) plus some organic reactions from Lurmann et al. (1986). It has been updated and expanded by von Glasow and Crutzen (2004) to include a better description of the oxidation of dimethylsulfide (DMS). Iodine chemistry was significantly improved by Pechtl et al. (2006, 2007). Further updates to the chemical mechanism were done by Sommariva and von Glasow (2012). The current mechanism is provided in the model manual (tables in Appendix D). In the following, the term aqueous phase is used as generic term for sub-cloud aerosol, interstitial aerosol (i.e. non-activated aerosol particles in cloudy layers), and cloud particles. Aqueous chemistry is not computed above the top of the boundary layer (i.e. the top of clouds, if present).

### 2.3.1 Gas phase and uptake

The prognostic equation for the concentration of a gas phase chemical species $c_{\mathrm{g}}$ (amount per air volume) including subsidence, turbulent exchange, deposition on the ocean surface, chemical production and destruction, emission and exchange with the aqueous phases is:

$$\frac{\partial c_{\mathrm{g}}}{\partial t} = -w\frac{\partial c_{\mathrm{g}}}{\partial z} + \frac{\partial}{\partial z}\left(K_{\mathrm{h}}\rho\frac{\partial c_{\mathrm{g}}/\rho}{\partial z}\right) + P - Sc_{\mathrm{g}} + E - Dc_{\mathrm{g}} - \sum_{i=1}^{n_{\mathrm{kc}}}\left[\overline{k_{t,i}}\left(w_{\mathrm{l},i}c_{\mathrm{g}} - \frac{c_{\mathrm{a},i}}{H_{\mathrm{s}}^{cc}}\right)\right]. \tag{10}$$

Again subsidence is the first term on the right and is included only in runs with clouds, otherwise $w = 0$. The second term on the right hand side of equation (10) describe the vertical turbulent mixing. $P$ and $S$ are chemical production and sink (i.e. loss) terms, respectively. The emission $E$ as well as dry deposition $D$ are effective only in the lowermost model layer. The calculation of the dry deposition velocity $v_{\mathrm{g}}^{\mathrm{dry}}$, that is needed for the determination of $D$, is explained in Sect. 2.3.6. Note that both $E$ and $D$ are not inserted as fluxes in equation (10). Instead, the actual fluxes have to be divided by the thickness of the lowermost model layer to yield $D$ and $E$. The last term in equation (10) describes the transport between the gas phase and the aqueous phases according to the formulation by Schwartz (1986) (see also Sander, 1999). In this term, $n_{kc}$ is the number of aqueous classes (see Sect. 2.3.2 ), $H_{\mathrm{s}}^{cc}$ is the dimensionless Henry constant obtained by $H_{\mathrm{s}}^{cc} = H_{\mathrm{s}}^{cp}RT$, where $H_{\mathrm{s}}^{cp}$ is in $\mathrm{mol\,m^{-3}\,Pa^{-1}}$, and $w_{\mathrm{l},i}$ is the dimensionless liquid water content ($V_{\mathrm{aq}}/V_{\mathrm{air}}$) of bin $i$.

For a single particle, the mass transfer coefficient $k_{\mathrm{t}}$ is defined as:

$$k_{\mathrm{t}} = \left( \frac{r^2}{3D_{\mathrm{g}}} + \frac{4r}{3\bar{v}\alpha} \right)^{-1} \tag{11}$$

with the particle radius $r$, the mean molecular speed $\bar{v} = \sqrt{8RT/(M\pi)}$ ($M$ is the molar mass), the accommodation coefficient $\alpha$, and the gas phase diffusion coefficient $D_{\mathrm{g}}$. $D_{\mathrm{g}}$ is approximated using the mean free path length $\lambda$ as $D_{\mathrm{g}} = \lambda\bar{v}/3$ (e.g. Gombosi (1994), p. 125).

Chameides (1984) points out that the time needed to establish equilibrium between the gas and aqueous phase differs greatly for individual species and that soluble species never reach equilibrium in cloud droplets, emphasizing the importance of describing phase transfer in the kinetic form that is used here. Audiffren et al. (1998) and Chaumerliac et al. (2000) point out that for reactive species like $H_2O_2$, the use of the Henry equilibrium assumption instead of the detailed description of mass transfer in the kinetic form that is used here would lead to significant errors in cloud droplet concentrations.

Ambient particle populations are never monodisperse, i.e. one has to account for particle with different radii. The transfer coefficient $\overline{k_{\mathrm{t}}}$ for a particle population is given by the integral:

$$\overline{k_{\mathrm{t}}} = \frac{4\pi}{3w_l} \int\limits_{\lg r_{\min}}^{\lg r_{\max}} \left( \frac{r^2}{3D_{\mathrm{g}}} + \frac{4r}{3\bar{v}\alpha} \right)^{-1} r^3 \frac{\partial N}{\partial \lg r} d\lg r, \tag{12}$$

where the size distribution function $\partial N/\partial \lg r$ depends on the type of aerosol chosen.

### 2.3.2   Aqueous phase

Aqueous chemistry is calculated in four bins (see Figure 2): deliquescent aerosol particles with a dry radius less than 0.5 µm are included in the "sulfate aerosol" bin #1, whereas deliquescent particles with a dry aerosol radius greater than 0.5 µm are in the "sea salt aerosol" bin #2. Although the composition of the particles changes over time, the terms "sulfate" and "sea salt" aerosol are used to describe the origin of the particles. The particles get internally mixed by exchange with the gas phase but, as mentioned earlier, not by particle coagulation. Depending on the type of aerosol relevant to the study, various initial
compositions of the aerosol bins may be chosen.

When the total particle radius exceeds the dry particle radius by a factor of 10, i.e. when the total particle volume is 1000 times greater than the dry aerosol volume, the particle and its associated chemical species are moved to the corresponding sea salt or sulfate-derived cloud particle class (#3 and #4, respectively). This threshold roughly coincides with the critical radius derived from the Köhler equation (see Eq. 8). Conversely, when particles shrink, they are redistributed from the droplet to the
aerosol bins.

Therefore in a cloud-free layer there are two ($n_{kc} = 2$) aqueous chemistry bins (sulfate and sea salt aerosol) and in a cloudy layer two cloud droplet (sulfate and sea salt derived) and two interstitial aerosol (sulfate and sea salt) bins, giving a total of four ($n_{kc} = 4$) aqueous chemistry bins. In each of these bins the following prognostic equation is solved for each chemical species $c_{\mathrm{a},i}$ (amount per air volume), where the index $i$ stands for the i-th aqueous bin:

$$\frac{\partial c_{a,i}}{\partial t} = -w\frac{\partial c_{a,i}}{\partial z} + \frac{\partial}{\partial z}\left(K_h\rho\frac{\partial c_{a,i}/\rho}{\partial z}\right) + P - Sc_{a,i} + E - D + P_{pc} + \overline{k_{t,i}}\left(w_{l,i}c_g - \frac{c_{a,i}}{H_s^{cc}}\right) \tag{13}$$

The individual terms have similar meanings as in equation (10). The calculation of the sedimentation velocity $v_{a,i}^{dry}$, that is needed for the calculation of the dry deposition $D$, is explained in Sect. 2.3.6. The additional term $P_{pc}$ accounts for the transport of chemical species from the aerosol to the cloud droplet regimes and vice versa when droplets are formed or when they evaporate, i.e. when particles move along the Köhler curve and get activated or unactivated. If only phase transfer is considered, equation (13) reduces in steady state conditions ($\partial c_{a,i}/\partial t = 0$) to the Henry equilibrium $c_{a,i} = w_{l,i}c_g H_s^{cc}$.

The concentration of $H^+$ ions is calculated like any other species, i.e. no further assumptions are made. The charge balance is satisfied implicitly.

### 2.3.3 Hysteresis of particle activation

Cloud-processing, i.e. the change of aerosol mass due to uptake of gases, is included based on the model of Bott (1999b).

It has been observed in many laboratory experiments that soluble aerosol remains in a highly concentrated metastable aqueous state when they are dried below their deliquescence humidity. Only when they reach the crystallisation humidity they can be regarded as "dry". This effect is called the hysteresis effect. For $NaCl$ the crystallisation point is about 45 % relative humidity (Shaw and Rood (1990), Tang (1997), Pruppacher and Klett (1997), and Lee and Hsu (2000)).

The crystallisation humidity for many mixed aerosol particles containing sulfate or nitrate is below 40 % relative humidity (Seinfeld and Pandis (2016) and references therein), implying that aerosol particles that already had been involved in cloud cycles will also be in an aqueous metastable state. Therefore most soluble aerosol particles will be present in the atmosphere as metastable aqueous particles below their deliquescence humidity. If the humidity drops below the crystallisation humidity, these particles can only reactivate when the deliquescence humidity is reached.

### 2.3.4 Accounting for the chemical activity

Aerosol particles are usually highly concentrated solutions. Laboratory measurements show that $NaCl$ molalities can be in excess of $10\ mol/kg$ (Tang, 1997) implying high ionic strengths. Therefore, it is necessary to account for deviations from ideal behaviour by including activity coefficients. The Pitzer formalism (Pitzer, 1991) is used to calculate the activity coefficients for the actual composition of each aqueous size bin. The implementation by Luo (Luo et al. (1995) and pers. comm. 1999) is used in MISTRA. It computes the activity of 7 main ions ($H^+$, $NH_4^+$, $Na^+$, $HSO_4^-$, $SO_4^{2-}$, $NO_3^-$, and $Cl^-$). The activities of 15 other ions are scaled on the previous ones based on the results from Liang and Jacobson (1999) and Chameides and Stelson (1992).

### 2.3.5 Photolysis

Here an overview of the calculation of the photolysis rates is given, for a detailed description see the model manual, Chapter 5. Photolysis is calculated online using the method of Landgraf and Crutzen (1998). The photolysis rate constant (or photo dissociation coefficient) $J_X$ for a gas $X$ can be calculated from the spectral actinic flux $F(\lambda)$ via the integral:

$$J_X = \int_I \sigma_X(\lambda)\phi_X(\lambda)F(\lambda)d\lambda \tag{14}$$

where $\lambda$ is the wavelength, $\sigma_X$ the absorption cross section, $\phi_X$ the quantum yield and $I$ the photochemically active spectral interval. If the integral in equation (14) were approximated with a sum, the number of wavelength intervals needed for an accurate approximation of the integral would be in the order of 100, which would lead to excessive computing times. Landgraf and Crutzen (1998) suggested a method using only 8 spectral intervals approximating (14) by:

$$J_X \approx \sum_{i=1}^{8} J_{i,X}^a \delta_i \tag{15}$$

where $J_{i,X}^a$ is the photolysis rate constant for a purely absorbing atmosphere. The factor $\delta_i$:

$$\delta_i = \frac{F(\lambda_i)}{F^a(\lambda_i)} \tag{16}$$

describes the effect of scattering by air molecules, aerosol and cloud particles. $F^a(\lambda_i)$ is the actinic flux of a purely absorbing atmosphere. The factor $\delta_i$ is calculated online for one wavelength for each interval, while the $J_{i,x}^a$ are pre-calculated with a fine spectral resolution and are approximated during runtime from lookup tables or by using polynomials. The advantage of this procedure is that the fine absorption structures that are present in $\sigma_X$ and $\phi_X$ are considered and only Rayleigh and cloud scattering, included in $F(\lambda_i)$, are treated with a coarse spectral resolution, which is justified.

For the calculation of the actinic fluxes, a four stream radiation code is used in addition to the two stream radiation code used for the determination of the net radiative flux density $E_n$, because different spectral resolutions and accuracies are needed for these different purposes. Based on the findings of Ruggaber et al. (1997), photolysis rates inside aqueous particles are increased by a factor of two to account for the actinic flux enhancement inside the particles due to multiple scattering.

### 2.3.6 Emission and deposition

The emission of gases is accounted for in the model, either with constant emission fluxes (for instance, for DMS and $NH_3$ emitted from the sea surface), or with scenarios of emission variable in time (see for instance the example run based on the study of Joyce et al. (2014)).

Sea salt particles are emitted by bursting bubbles at the sea surface (e.g. Woodcock et al. (1953), Pruppacher and Klett (1997)). The parameterisations of Monahan et al. (1986) and Smith et al. (1993) are implemented to estimates the flux of particles. The former is advised for small to moderate wind speeds, while the latter has to be used for high wind speeds.

The dry deposition velocity for gases $v_g^{\text{dry}}$ at the sea surface is calculated using the resistance model described by Wesely (1989):

$$v_g^{\text{dry}} = \frac{1}{r_a + r_b + r_c}.$$

(17)

The aerodynamic resistance $r_a$ is calculated using:

$$r_a = \frac{1}{\kappa u_*}\left[\ln\left(\frac{z}{z_0}\right) + \Phi_s(z, L_{\text{MO}})\right],$$

(18)

with the friction velocity $u_*$, the von Kármán constant $\kappa = 0.4$, and the stability function $\Phi_s$ which depends on the Monin-Obukhov length $L_{\text{MO}}$, the roughness length $z_0$ and a reference height $z$. The quasi-laminar layer resistance $r_b$ is parameterised for gases as:

$$r_b = \frac{5Sc^{2/3}}{u_*}.$$

(19)

The Schmidt number $Sc$ can be written as $Sc = \nu/D_{\text{g}}$ with the kinematic viscosity of air $\nu$ and the gas diffusion coefficient $D_{\text{g}}$ as in Eq. 11. The surface resistance $r_c$ is calculated using the formula by Seinfeld and Pandis (2016) (their equation (19.30)):

$$r_c = \frac{2.54 \times 10^4}{H^* T u_*},$$

(20)

with the effective Henry constant $H^*$.

The dry deposition velocity of particles $v_{a,i}^{\text{dry}}$ is calculated after Seinfeld and Pandis (2016):

$$v_{a,i}^{\text{dry}} = \begin{cases} \frac{1}{r_a + r_b + r_a r_b w_t} + w_t & \text{lowest model layer} \\ w_t & \text{rest of model domain.} \end{cases}$$

(21)

where the quasi-laminar resistance $r_b$ is parameterised for particles as:

$$r_b = \frac{1}{u_*(Sc^{-2/3} + 10^{-3/St})}.$$

(22)

The Stokes number $St$ can be written as $St = w_t u_*^2/(g\nu)$ where $g$ is the gravitational acceleration. The particle sedimentation velocity $w_t$ is calculated in the microphysical module assuming Stokes flow and taking into account the Cunningham slip flow correction for particles with $r < 10\mu$m and after Beard for larger particles (see Pruppacher and Klett, 1997).

Finally, the dry deposition term D is calculated as:

$$D = \exp\left(-\Delta t/h \times v_g^{\text{dry}}\right)$$

(23)

where $\Delta t$ is the model time step, and $h$ is the height of the lowermost model layer.

### 2.3.7 Nucleation

A module computing the nucleation process was implemented in MISTRA by Pechtl et al. (2006). Only a brief overview is given here, while a comprehensive description is given in the model manual (Chapter 4). The nucleation module developed by Pechtl et al. (2006) includes both ternary sulfuric acid-ammonia-water ($H_2SO_4-NH_3-H_2O$) nucleation, and homomolecular homogeneous OIO nucleation. The former is explicitly calculated as a function of $H_2SO_4$ and $NH_3$ concentrations, relative humidity, and temperature following the work by Napari et al. (2002). The latter is parameterised following Burkholder et al. (2004). Each process can be activated or not independently (see Table 1), and lead to the computation of "real" nucleation rates. In a second step, the "apparent" nucleation rate is computed after the work of Kerminen and Kulmala (2002) and Kerminen et al. (2004).

The nucleated particles computed in this module can then be integrated in the model, with three possible options: (i) no coupling, (ii) coupling with the microphysics without feedback on chemistry, and (iii) coupling with microphysics and chemistry (see Table 1).

## 3 Technical description

### 3.1 Namelist settings

**General configuration switches**

Table 1 presents the switches available to define the model configuration.

**Initialisation and run settings**

Initial atmospheric conditions are set in the namelist with the parameters presented in Table 2. All these parameters have default values, even if most of them are expected to be redefined by the user to match the simulated atmosphere. Standard settings (timing and geography, run duration) are straightforward and are not detailed hereafter. Typical run duration covers a few hours to a few days. Longer run duration is sometimes necessary for model spin up. The restart option of the model allows a single spin up run to initialise the model, and perform a sensitivity analysis from that stage, for instance. In addition to these, surface settings are detailed in Table 3.

**Special runs setting**

When a specific run requires multiple adjustments in various parts of the code that were not already including namelist options, a single general switch might be used instead of defining several new namelist entries for each parameterisation that require

**Table 1.** General configuration switches in MISTRA-v9.0

| Switch name | Description |
|---|---|
| rst | defines if the model is restarted (true) or not (false, default setting). |
| mic | this switch is used to turn on (true, default setting) or off (false) the 2D microphysical distribution |
| chem | this switch is used to turn on (true, default setting) or off (false) the whole chemistry module. If turned off, only the physics and microphysics is activated. |
| halo | activate (default setting, true) or unactivate (false) the chemical reactions involving halogen species (Cl, Br, and I) |
| iod | activate (default setting, true) or unactivate (false) the chemical reactions involving iodine species |
| nuc | activate (true) or unactivate (false, default setting) the nucleation module. |
| Napari | activate (true, default setting) or unactivate (false) the ternary $H_2SO_4$-$H_2O$-$NH_3$ nucleation |
| Lovejoy | activate (true, default setting) or unactivate (false) the OIO homogeneous nucleation |
| ifeed | nucleation feedback over background particles (0=no feedback, default setting; 1=with feedback, 2=partial feedback for microphysics only, see Chapter 4 of the manual for a complete description of the nucleation module). |
| box | use the box version (0D: true) or the whole 1D version (false, default setting) of the chemistry module. |
| BL_box | define whether the box represents a single layer extracted from the 1D model (BL_box = false, default setting) or an average of the whole boundary layer (BL_box = true). |
| nlevbox | index of the designated layer if BL_box=false |
| z_box | height of the boundary layer represented by the box if BL_box=true |
| chamber | use the chamber version (true) or the whole 1D version (false, default setting) of the chemistry module. |
| binout | request binary output files (true) or not (false, default setting). |
| netcdf | request netCDF[a] output files (true, default setting) or not (false). |

[a] network Common Data Form, see https://www.unidata.ucar.edu/software/netcdf/ (last accessed 21/11/2021).

special settings. An example of such global switch for special configuration is given with the lpJoyce14bc switch, used to activate all relevant parts of code to reproduce the base case of Joyce et al. (2014) study (particle distribution and composition, gas and particle emission scenario, special formulation of accommodation coefficient for $N_2O_5$, and of gas dry deposition, etc.). Similarly, switches lpBuys13_0D and lpBuxmann15alph are used to reproduce all relevant settings of the studies of Buys et al. (2013, 0D case) and Buxmann et al. (2015, alpha case), respectively.

### 3.2 How to run MISTRA

#### 3.2.1 Get the code

The model is provided on GitHub on the following repository: https://github.com/Mistra-UEA/Mistra. It is released under the European Union Public Licence (EUPL) v1.1, which permits free commercial and private use and unrestricted distribution, but requires that future developments of MISTRA are shared under the same licence. The version of KPP adapted for MISTRA is provided along with the distribution, and is released under its own licence.

**Table 2.** Namelist settings for model initialisation of MISTRA-v9.0

| Parameter name | Unit | Description |
| --- | --- | --- |
| detamin | m | constant vertical grid spacing for the lowest 100 prognostic layers (default = 10 m). |
| etaw1 | m | height of the highest prognostic layer (50 layers with exponentially increasing thickness on top of the 100 layers with constant height). Default is 2000 m. |
| rnw0 | µm | minimum dry particle radius in the 2D particle spectrum. |
| rnw1 | µm | maximum dry particle radius in the 2D particle spectrum. |
| rw0 | µm | minimum total particle radius in the 2D particle spectrum. |
| rw1 | µm | maximum total particle radius in the 2D particle spectrum. |
| iaertyp | – | aerosol distribution: 1 for urban aerosol, 2 for rural aerosol, 3 for marine aerosol (default setting). |
| rp0 | Pa | pressure at ground/sea level. |
| zinv | m | inversion height. It must be lower than the highest prognostic layer. |
| dtinv | K | temperature jump at inversion level. |
| ug | $\mathrm{m\,s^{-1}}$ | geostrophic wind speed in x direction. |
| vg | $\mathrm{m\,s^{-1}}$ | geostrophic wind speed in y direction. |
| nuvProfOpt | – | option for vertical profile of geostrophic wind speed components. By default, the same values are applied to the whole atmospheric column, apart from the 4 lowest layers where wind components are reduced to 75 %, 50 %, 25 %, and 0 % at ground/sea level. Alternatively, using nuvProfOpt, other specific profiles can be defined. Currently, only one alternative profile for geostrophic wind components is proposed: if nuvProfOpt=3 is selected, the geostrophic wind is constant above the inversion level, and linearly decreases in the whole MBL to reach zero at ground/sea level. |
| wmin | $\mathrm{m\,s^{-1}}$ | minimum subsidence speed (default is $0\,\mathrm{m\,s^{-1}}$). |
| wmax | $\mathrm{m\,s^{-1}}$ | maximum (in negative values) subsidence speed. |
| nwProfOpt | – | profile option for subsidence. See appendix 1 for a description. |
| xm1w | $\mathrm{kg\,kg^{-1}}$ | moisture content in the MBL |
| xm1i | $\mathrm{kg\,kg^{-1}}$ | moisture content in the FT (i.e. above inversion level). |
| rhMaxBL | 1 | maximum relative humidity (default=1) in the MBL (additional constrain to xm1w parameter). |
| rhMaxFT | 1 | maximum relative humidity in the FT (additional constrain to xm1i parameter). |
| cGasListFile | | names of user gas files for non radical species. |
| cRadListFile | | names of user gas files for radical species. |
| scaleO3_m | D.U. | ozone column, to scale the photolysis rates computed. It has no effect on the radiation calculation. |

### 3.2.2 System requirements and installation

A Fortran compiler is required to compile the model code. During the recent development stages, MISTRA has been regularly compiled using either GNU Fortran (gfortran) or Intel Fortran (ifort). New users are advised to choose one of those compilers.

**Table 3.** Namelist settings for surface initialisation and parameterisation in MISTRA-v9.0

| Parameter name | Unit | Description |
|---|---|---|
| isurf | | sets the type of surface, 0 for ocean or snow (default setting), 1 for layered soil. |
| tw | K | initial surface temperature. |
| ltwcst | | constant (true, default setting) or time-varying (false) surface temperature. |
| ntwopt | | scenario number for time varying surface temperature (if ltwcst = true). |
| jpAlbedoOpt | | surface albedo option[a]: 0 for ocean surface (default setting: albedo = 0.05), 1 for snow surface (albedo = 0.8). |
| z0 | m | roughness length at the surface. |
| lpmona | | aerosol source after Monahan et al. (1986) (small to moderate wind speed) |
| lpsmith | | aerosol source after Smith et al. (1993) (high wind speed) |

[a] Currently, constant albedo is used over the 6 solar wavelength bands. Alternative choice with varying albedo could be implemented with this namelist option.

The implementation of KPP output files into MISTRA is done with bash and csh scripts, thus any change in the chemical mechanism will require these shells.

Plotting scripts provided as example are written for Ferret (http://ferret.pmel.noaa.gov/Ferret/, last accessed 04/11/2021) and NCL (NCAR, 2019), but neither are necessary to run the model. Only KPP needs to be installed on the user system and the instructions to do so are in the readme file of the KPP distribution package. Preprocessed files using the current chemical
mechanism are provided in the distribution, so the installation of KPP can be skipped until the user needs to modify the chemical mechanism.

### 3.2.3 Prepare the chemical mechanism files

This section can be skipped if no change is applied to the current chemistry mechanism. All files related to the chemical mechanism are contained in the subdirectory ./src/mech The chemical mechanism, written with the formalism of KPP, is
360 contained in two main files: master_gas.eqn for gas phase reactions, and master_aqueous.eqn for the liquid phase mechanism. For convenience, all necessary steps to prepare the equation files for KPP, run KPP, and adapt the resulting output from KPP for MISTRA have been set up in a Makefile, so that the user simply has to run make in the ./src/mech directory to proceed. The resulting files are copied to the main source directory.

### 3.2.4 Compile the model

In ./src, after ensuring that the Makefile refers to the correct Fortran compiler, and links to the appropriate netCDF libraries, compile the model running make. The resulting executable file is mistra.

### 3.2.5 Set a namelist and initial chemical species concentration

As presented in details in Sect. 3.1, the namelist file allows the user to configure the model, by setting the main options and initialisation values. Several namelists are provided in the distribution and can be used as starting points to define new ones

corresponding to the user requirements. The set of initial concentration, and emission of gas phase species can be set in a tab-separated table, whose name has to be specified in the namelist. If no file is specified, the `./src/mech/gas_species.csv` file is used by default.

### 3.2.6  Set a param file and run

The param file allows the user to specify the namelist to use for the run, and to define the paths to the input, output and mechanism directories. For most cases, these directories will be the default ones (`./input`, `./output`, and `./src/mech`), and only the namelist name should be specified. Several param files are provided as example. This script is in charge of creating the subdirectory for output (which will be named the same as the param file name), and launch the model.

## 4  Consistency with previous versions

In this section, we present a series of example runs that have been performed to evaluate the model. All the examples provided here reproduce the settings of previous studies carried out with previous versions of MISTRA. For that purpose, several namelists have been introduced to hold all relevant parameters in order to reproduce the same simulation scenarios as in the original publications. These namelists, as well as the scripts used to produce the plots presented here, are available in the MISTRA repository.

### 4.1  Meteorology and microphysics

#### 4.1.1  Comparison with 1996 version: LWC, TKE and 2D spectrum

The first example focuses on the physical and microphysical aspects of the model. For this purpose, the chemistry is switched off. The initialisation settings are identical to those of the original paper from Bott et al. (1996), and are provided in the namelist `BTZ96`. Some model changes have been maintained for this comparison, even if this leads to differences to the original version. For instance, the number of bins in the 2D particle spectrum is set to $70 \times 70$ in MISTRA-v9.0 while it was of $40 \times 50$ in the version of Bott et al. (1996). However, we adjusted the minimum and maximum particle radius values so that the resolution is nearly identical in both versions.

This simulation reproduces conditions over the North Sea, for 3 simulated days (2 are shown, the first one is used as model spin up) centered on the 22 July. The radiative code used by Bott et al. (1996) has been updated from PIFM1 to PIFM2 by Loughlin et al. (1997), and the simulation settings were the same in both papers. For this reason, we compare figures from the study of Loughlin et al. (1997) with the current output of MISTRA.

Both Fig. 3 and Fig. 4 show very similar model output between the 1996 version of MISTRA and the current version. The runs are similar, qualitatively and quantitatively (the maximum LWC is 6 % higher, the maximum TKE is 2.5 % higher in MISTRA-v9.0 than in the 1996 version), without changing the findings and conclusions of the original study.

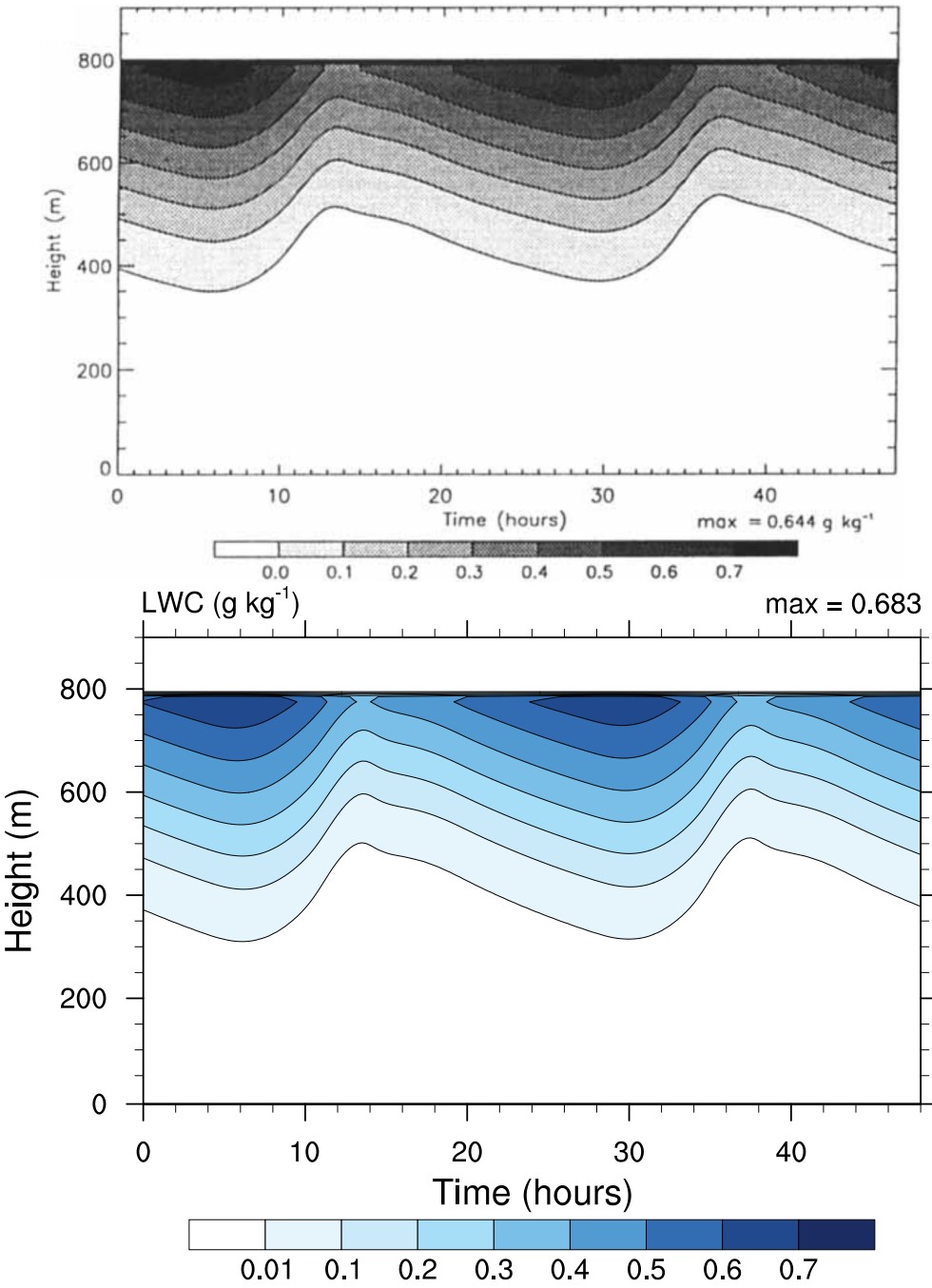

**Figure 3.** Contour plot of cloud water content (in $g\,kg^{-1}$) as a function of height and simulation time. Top: study from Loughlin et al. (1997, Fig. 1a). Bottom: MISTRA-v9.0. For both, the simulation settings are identical to those in Bott et al. (1996). Note the minimum contour level is set to 0.01 in both panels, but was displayed incorrectly in the original figure. Top panel reproduced with permission from Loughlin et al. (1997, Fig. 1a). Copyright (1997) by John Wiley and Sons.

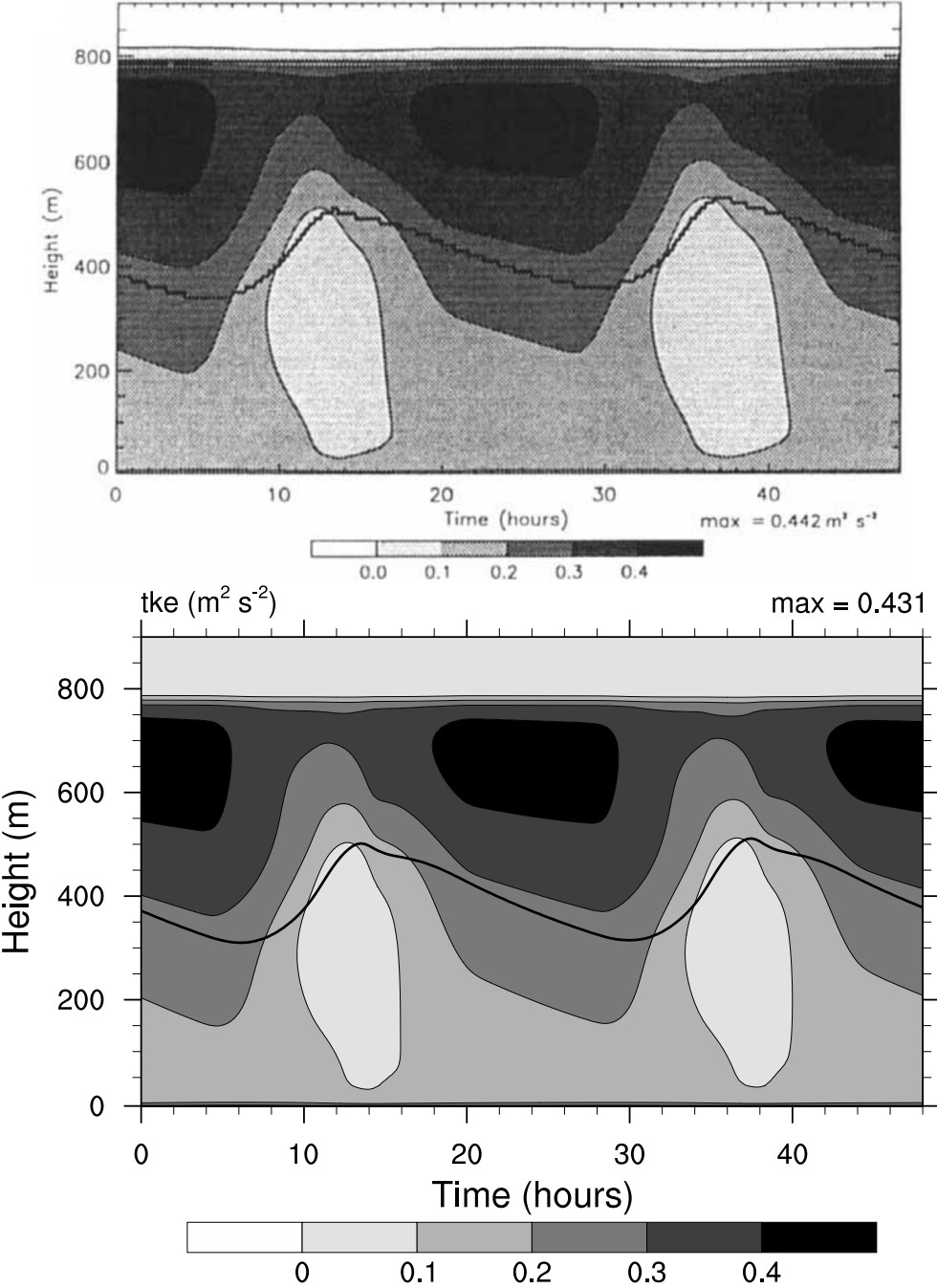

**Figure 4.** Contour plot of turbulent kinetic energy (in $m^2\,s^{-2}$ as a function of height and simulation time. The thick line shows the bottom of cloud, defined as LWC dropping below $0.01\ g\,kg^{-1}$. Top: study from Loughlin et al. (1997, Fig. 4a). Bottom: MISTRA-v9.0. For both, the simulation settings are identical to those in Bott et al. (1996). Top panel reproduced with permission from Loughlin et al. (1997, Fig. 4a). Copyright (1997) by John Wiley and Sons.

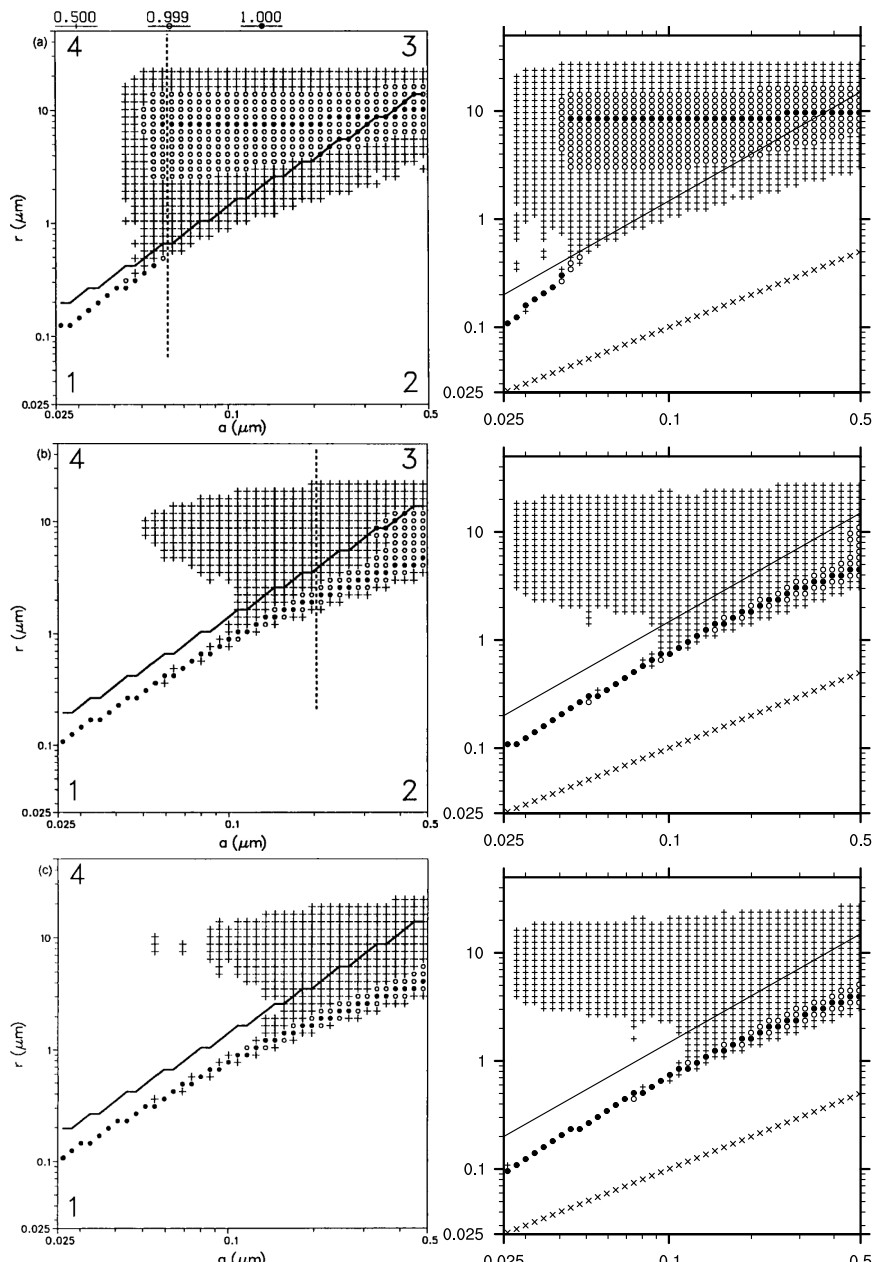

**Figure 5.** Distribution of particles in the 2D particle grid and at different heights in the cloud. This graph is built identical to Bott et al. (1996, Fig. 12): for each dry radius class (on x-axis), the total radius containing the maximum number of particles is marked with a filled circle; total radii containing 50 % to 99 % of the maximum are marked with open circles, and total radii containing 1 % to 50 % of the maximum are marked with plus signs. The 1:1 values (i.e. where total radius equal to dry radius) are represented with cross signs. The full line shows the activation radius as accounted for in MISTRA. From top to bottom row is top (785 m), middle (605 m), and bottom (555 m) of the cloud. Left column: version of 1996, panels are from Bott et al. (1996, Fig. 12). Right column: MISTRA-v9.0 Left column reproduced with permission from Bott et al. (1996, Fig. 12). Copyright (1996) by John Wiley and Sons.

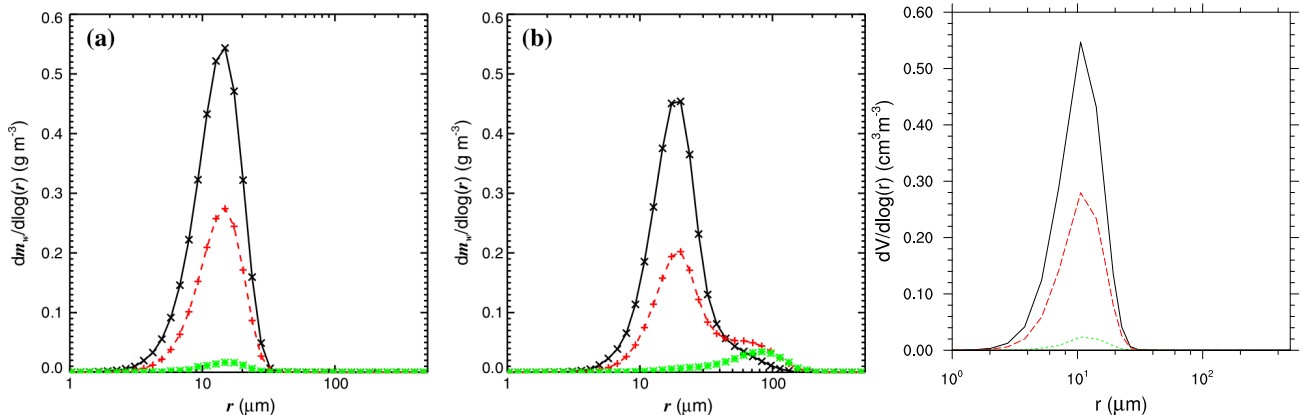

**Figure 6.** One dimensional distribution of particle mass as a function of radius. Left: no-chem-MISTRA without collision coalescence. Center: no-chem-MISTRA with collision-coalescence. Right: MISTRA-v9.0 (without collision coalescence implemented). Each panel shows the distribution in the top (black line), middle (red line), and bottom of cloud (green line). The left and central panels are from Bott (2020, Fig. 2, published under CC BY 4.0 licence). The simulation settings were taken from Bott (2020).

Bott et al. (1996) also shown the distribution of particles in the 2D grid, and we used the same graph format in Fig. 5.
Qualitatively, the two simulations are similar. MISTRA-v9.0 exhibits more particle growth for the smallest dry radius bins; however, this happens only for a minority of the particles ('plus' signs in Fig. 5 denotes bins where particle concentration is less than half the maximum particle concentration, for each dry radius class). As stated previously, the 1996 version of MISTRA used by Bott et al. (1996) included the first version of the radiative code PIFM1, now updated to PIFM2. The differences between both radiation schemes are likely the reason for this slightly different particle distributions observed for
small dry particle radius. This figure also highlights the microphysics properties and dynamics of particle within the cloud, with the activation of particles occurring when the supersaturation (not shown) is high enough, which is the case in the upper part of the cloud. Conversely, in the middle and bottom parts of the cloud most of the particles are found below the critical radius, even if some particles grow to above their respective critical radius, since the supersaturation is not high enough.

### 4.1.2 Impact of neglecting coalescence

As pointed out in the general presentation of the model (Sect.2.2), the collision-coalescence process is not accounted for in MISTRA-v9.0, which is a limitation of the model. The collision-coalescence process was implemented in a version of MISTRA without chemistry (Bott, 2000), hereafter referred to as MISTRA-coal-nochem for brevity. In a recent study, Bott (2020) used MISTRA-coal-nochem and compared the results with and without activating this process (Fig. 6). He showed that accounting for the collision-coalescencce process leads to significant differences in the particle distribution, with a bimodal spectrum
with particles larger than $40\,\mu m$ when collision-coalescence is activated (Fig. 6b) Conversely, when particles grow solely by diffusional uptake of water vapour, their size distribution remain in the 2 to $30\,\mu m$ range (Fig. 6b). We defined a namelist, named `Bott2020`, reproducing the same settings as in Bott (2020) to perform a further evaluation of MISTRA-v9.0 against

MISTRA-coal-nochem. The resulting one-dimensional particle distribution is presented in Fig. 6c, and shows similar results as compared to MISTRA-coal-nochem (Fig. 6a).

Despite the important differences in particle distribution when collision-coalescence process is included, this limitation in MISTRA-v9.0 is expected to have insignificant effect for simulations without clouds (non-activated particles only). Conversely, cloudy runs should be restricted to conditions where collision-coalescence is less important, i.e. cases where no or little drizzle formation would be expected. According to Duynkerke (1998), drizzle formation starts to be important when the cloud depth is greater than 300 m. Future development plans with MISTRA-v9.0 include a re-evaluation of the feasibility of including the
collision-coalescence process along with chemistry.

## 4.2   Chemistry in 1D simulations

A namelist reproducing the settings of the study by Joyce et al. (2014) is provided as `namelist.Joyce14bc`. In this study, MISTRA was used to simulate an urban pollution plume from Fairbanks, Alaska. The model was thus used in an alternative configuration, with surface covered by snow (with the relevant physical properties). An emission scenario of $NO_x$ ($NO+NO_2$)
was defined, and the evolution of gas and aqueous phase species was evaluated. In such configuration, the meteorological parameters have a strong influence over the stability of the atmosphere, thus in turn over the vertical exchange of chemical species. In Fig. 7, key meteorological variables are presented for both the original study and the new runs obtained with MISTRA-v9.0. As expected, there is excellent agreement between both versions, which shows that the recent code developments did not alter the model with regards to the plotted variables.

Figures. 8 and 9 show the comparison of gas and particle phase chemical species, between the original study of Joyce et al. (2014) and MISTRA-v9.0. Again, the plotted variables agree very well between both versions, with nearly identical results. In Fig. 9, the only exception is ammonium ($NH_4^+$, bottom right panel on each side) whose maximum concentration is decreased by 60 % in MISTRA-v9.0. The reason of this change was investigated, and we found that in the original run, the initialisation of a variable in the routine computing the gas-particle exchange rates was missing. This is now corrected in MISTRA-v9.0,
and explains the differences for $NH_4^+$.

## 4.3   Box and chamber model configurations

We present two additional configurations of MISTRA-v9.0, as an atmospheric box (0D) model (Buys et al., 2013), and in chamber configuration following the study of Buxmann et al. (2015). In both cases, we set namelists (`namelist.Buys13_0D` and `namelist.Buxmann15_alpha`, respectively) with the same settings as in the original publications.
Figures 10 and 11 compare model output between each model version, and show that minor differences exist but are very limited, and the results agree well qualitatively. This comes as a further demonstration of the good consistency of MISTRA-v9.0 with previous results.

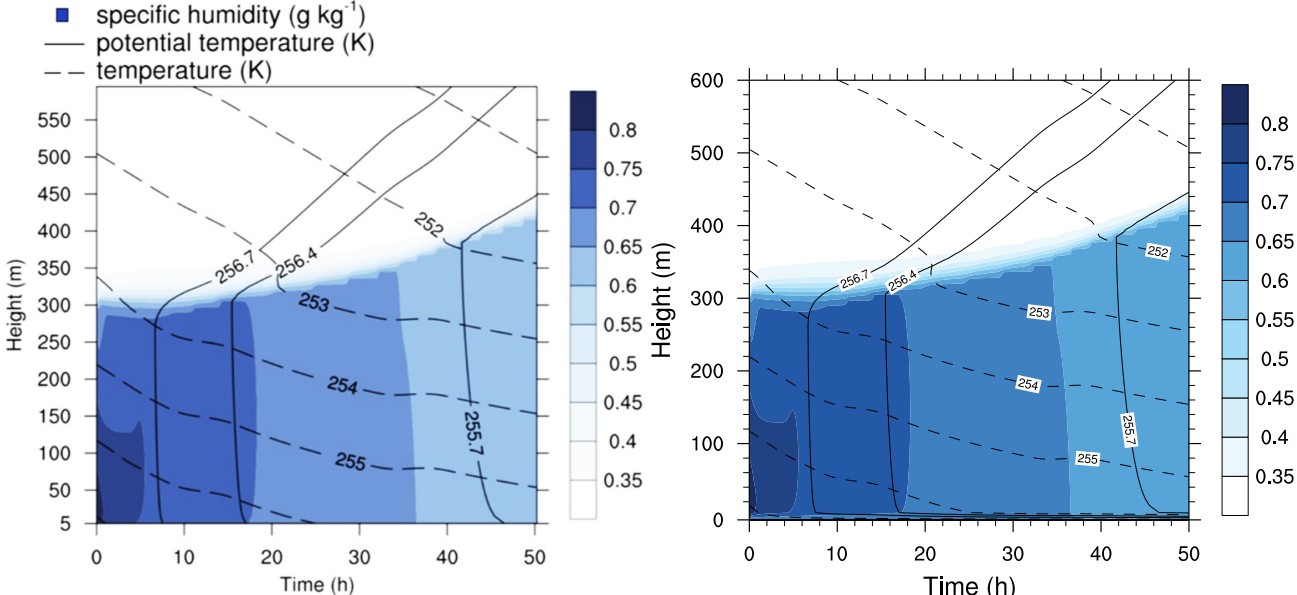

**Figure 7.** Contour plot of humidity, potential temperature and temperature in the run based on Joyce et al. (2014). Left panel: original study (Figure published under CC BY 3.0 licence). Right panel: MISTRA-v9.0.

## 5 Conclusions

We have presented the current version of the 0D/1D atmospheric chemistry model MISTRA-v9.0, released for the first time as an open-source, community model. MISTRA-v9.0 is a versatile model with a range of capabilities, from the study of status cloud microphysics, radiative forcing and turbulence, to the mutiphase atmospheric chemistry of the boundary layer. While its original purpose was only the study of cloud-free and cloudy marine boundary layer, MISTRA was successfully extended in previous studies to model other environments such as polar conditions and volcanic plumes. In this study, we updated the model code to comply with coding standards, and we compared current output of a range of test cases against previous studies with identical settings. Results obtained with MISTRA-v9.0 are consistent with the previous results even after 20 years of development. MISTRA-v9.0 is a powerful tool for atmospheric chemistry research purposes, now easier to use, and free to use under EUPL-v1.1 licence. Community input and development is welcome for MISTRA-v9.0.

*Code availability.* The code of the MISTRA-v9.0 model, the code of KPP-v2.2.3 tuned for MISTRA (referred to as v2.2.4), the additional example namelists and param files, and all NCL scripts developed to produce the Figures in this article are available on https://github. com/Mistra-UEA/Mistra. The archives of code releases are also available on Zenodo: https://doi.org/10.5281/zenodo.5110025 and http: //doi.org/10.5281/zenodo.5109913 for MISTRA-v9.0 and KPP-v2.2.4, respectively.

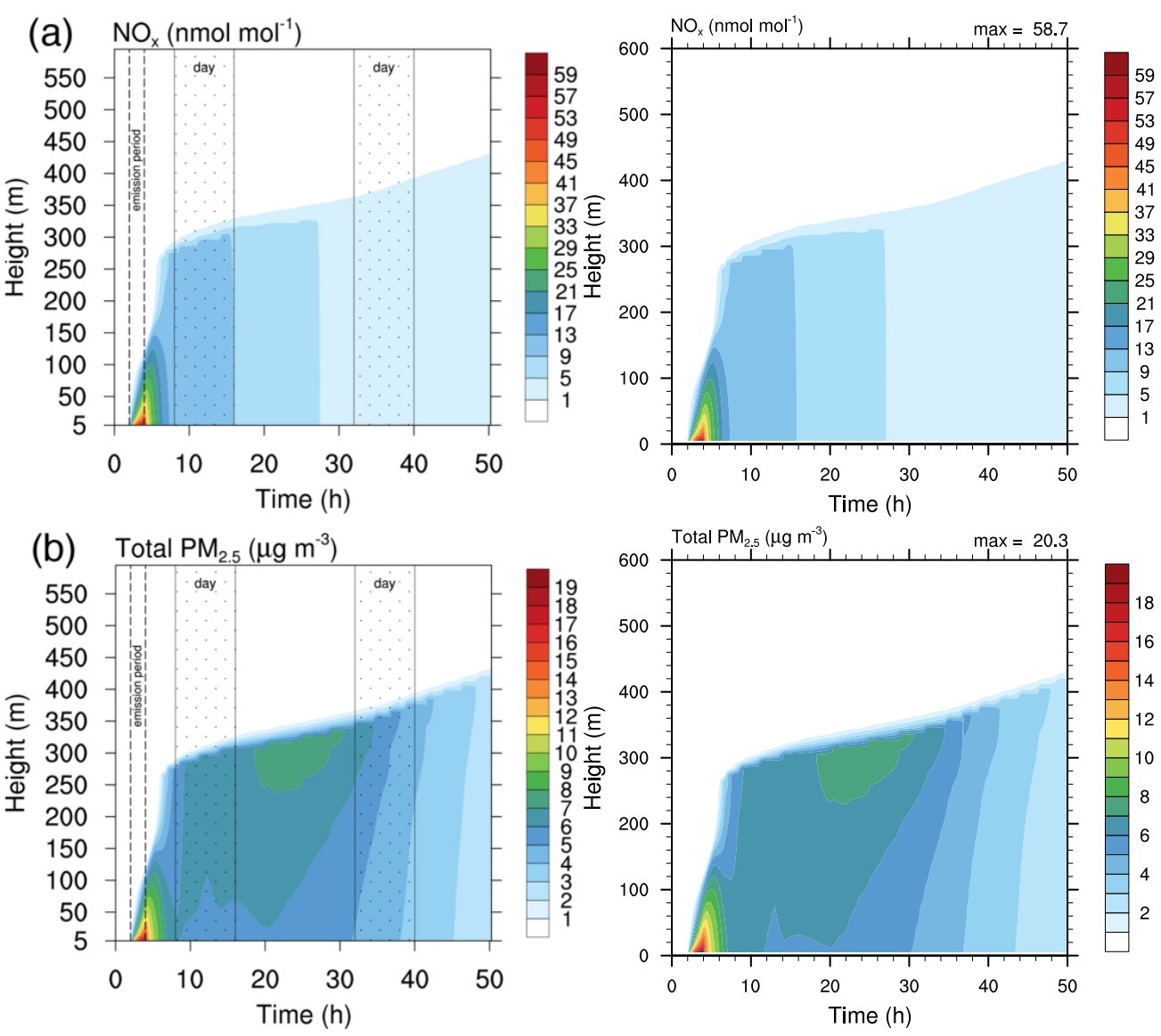

**Figure 8.** Contour plot of $NO_x$ and $PM_{2.5}$ in the run based on Joyce et al. (2014) study. Left panel: original study (Figure published under CC BY 3.0 licence). Right panel: MISTRA-v9.0. Scales are identical for both.

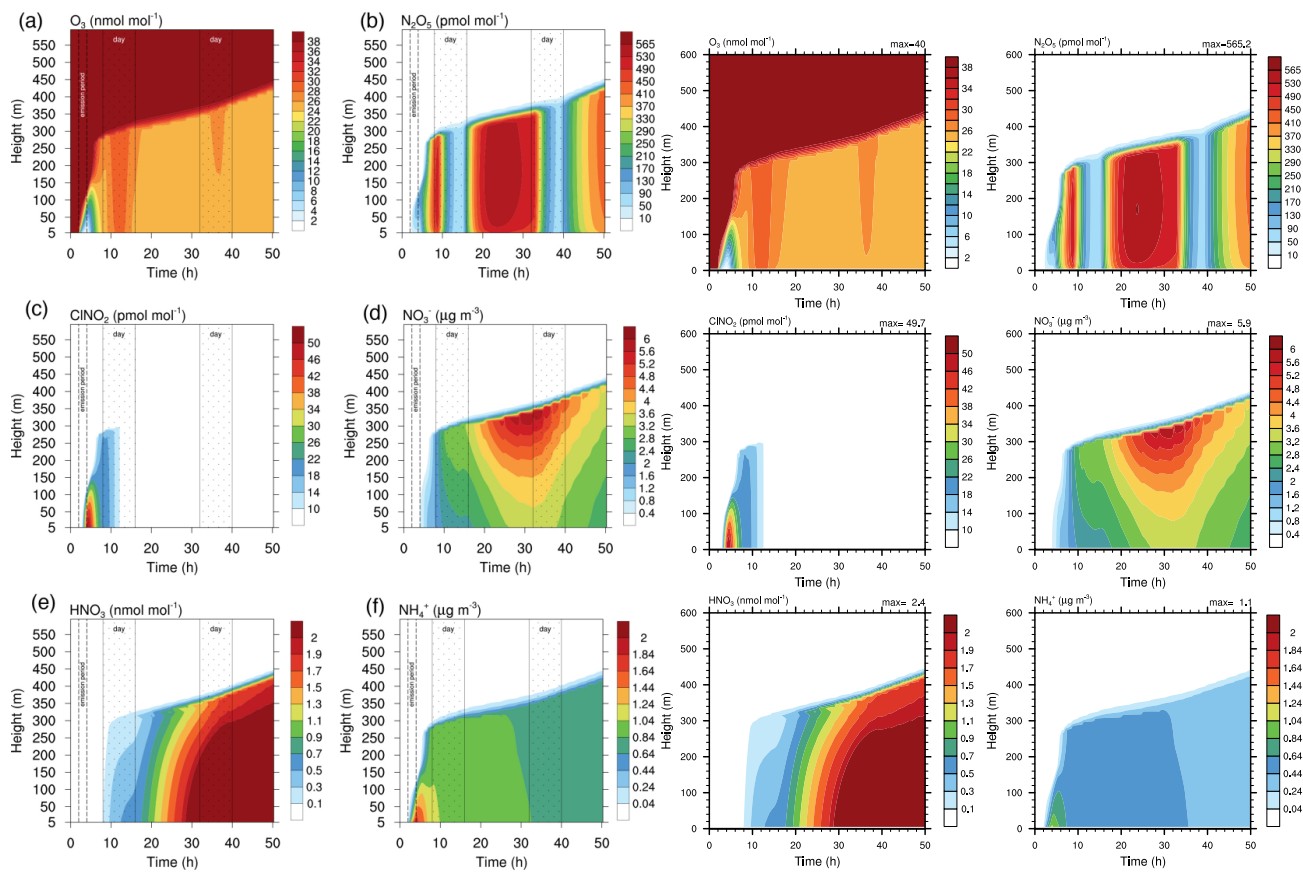

**Figure 9.** Contour plot of height versus time for 4 gases and 2 aqueous phase species in the run based on Joyce et al. (2014). Left side: original study (Figure published under CC BY 3.0 licence). Right side: MISTRA-v9.0. Scales are identical, except $NH_4^+$ where the scale for MISTRA-v9.0 is half that of the original paper.

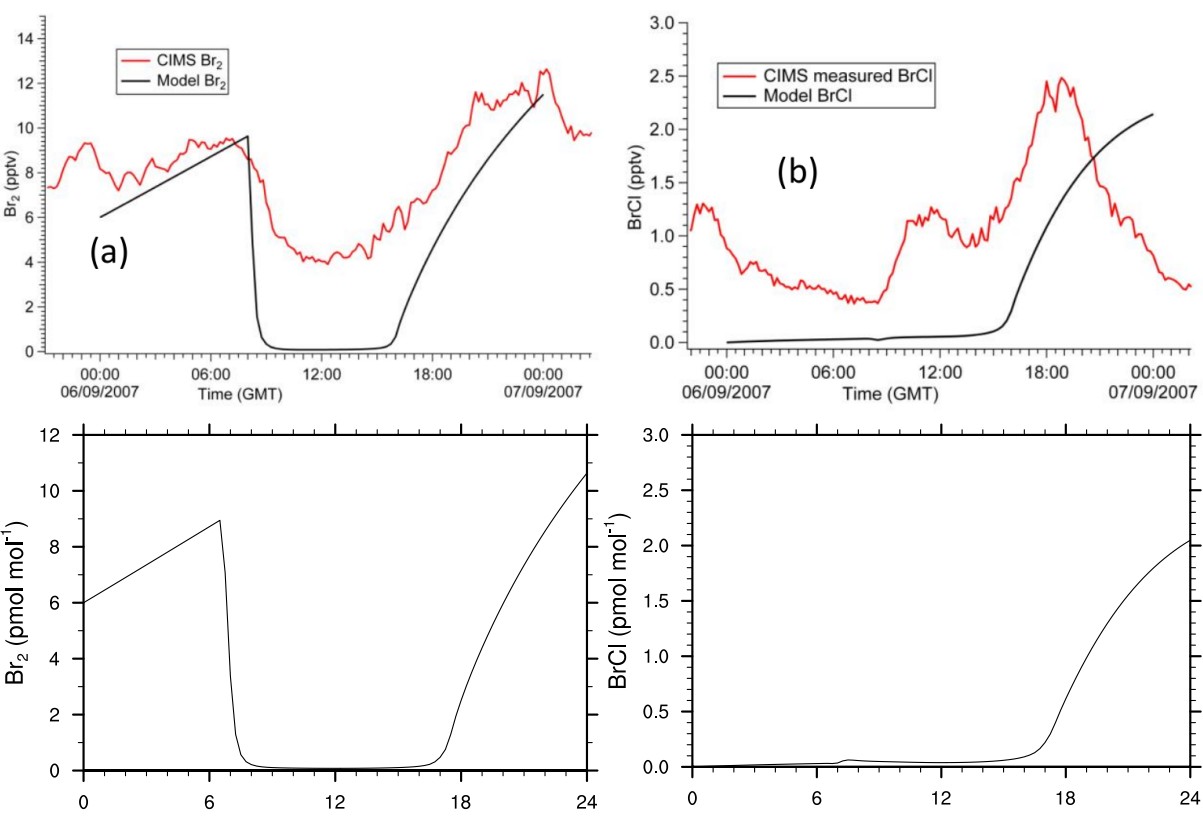

**Figure 10.** Gas phase concentration of $Br_2$ and $BrCl$ in a box model run based on the settings of Buys et al. (2013). Top row: original study (Figure 3 published under CC BY 3.0 licence). Bottom row: MISTRA-v9.0. Scales are identical.

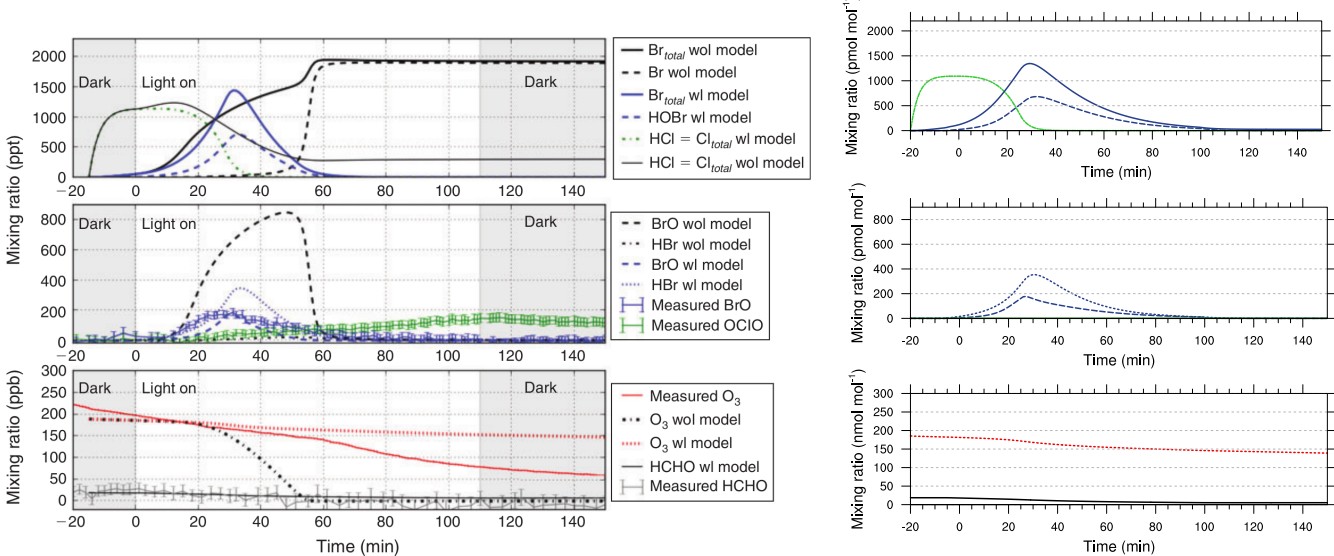

**Figure 11.** Gas phase concentration in a chamber model run based on the study of Buxmann et al. (2015) (alpha wl sensitivity experiment). Left side: original study. Right side: MISTRA-v9.0 with identical settings. Scales are identical. Left panel reproduced with permission from Buxmann et al. (2015, Fig. 5). Copyright (2015) by CSIRO Publishing.

## Appendix A:  List of symbols

| | | |
|---|---|---|
| $a$ | dry particle radius | m |
| $A$, $B$ | Kelvin effect, solute effect factors in the Köhler equation | m and 1 |
| $c_{a,i}$ | aqueous phase concentration in bin $i$ (per air volume) | $\mathrm{mol\,m^{-3}}$ |
| $c_g$ | gas phase concentration | $\mathrm{mol\,m^{-3}}$ |
| $c_p$ | specific heat of dry air at constant pressure | $1005\ \mathrm{J\,kg^{-1}\,K^{-1}}$ |
| $c_w$ | specific heat of water | $\mathrm{J\,kg^{-1}\,K^{-1}}$ |
| $C$ | condensation rate | $\mathrm{kg\,m^{-3}\,s^{-1}}$ |
| $D$ | dry deposition rate | $\mathrm{s^{-1}}$ |
| $D'_v$ | diffusivity of water vapour | $\mathrm{m^2\,s^{-1}}$ |
| $D_g$ | gas phase diffusion coefficient | $\mathrm{m^2\,s^{-1}}$ |
| $e$ | turbulence kinetic energy (TKE) | $\mathrm{m^2\,s^{-2}}$ |
| $E$ | emission of chemical species | $\mathrm{mol\,m^{-3}\,s^{-1}}$ |
| $E_n$ | net radiative flux density | $\mathrm{W\,m^{-2}}$ |
| $f(a,r)$ | aerosol or droplet particle number concentration | $\mathrm{m^{-3}}$ |
| $f_c$ | Coriolis parameter | $\mathrm{s^{-1}}$ |
| $F_d(a,r)$ | net radiative flux at the particle's surface | W |
| $g$ | gravitational acceleration | $9.80665\ \mathrm{m\,s^{-2}}$ |
| $G_h$, $G_m$ | stability functions for buoyancy and shear production (see also $S_h$, $S_m$) | 1 |
| $h$ | model layer height | m |
| $H^*$ | effective Henry constant | $\mathrm{mol\,m^{-3}\,Pa^{-1}}$ |
| $H_s^{cc}$ | dimensionless Henry's law solubility constant | 1 |
| $H_s^{cp}$ | Henry's law solubility constant | $\mathrm{mol\,m^{-3}\,Pa^{-1}}$ |
| $J_X$ | photolysis rate constant | $\mathrm{s^{-1}}$ |
| $k'$ | thermal conductivity of moist air | $\mathrm{W\,m^{-1}\,K^{-1}}$ |
| $k_t$ | mass transfer coefficient | $\mathrm{s^{-1}}$ |
| $\overline{k_t}$ | mean mass transfer coefficient including liquid water content | $\mathrm{s^{-1}}$ |
| $K_h$, $K_m$, $K_e$ | turbulent exchange coefficient for heat, momentum, and turbulence kinetic energy | $\mathrm{m^2\,s^{-1}}$ |
| $l$ | mixing length | m |
| $L$ | latent heat of condensation | $\mathrm{J\,kg^{-1}}$ |
| $L_{MO}$ | Monin-Obukhov length | m |
| $m_w$ | liquid water mass of the particle | kg |
| $M$ | molar mass | $\mathrm{kg\,mol^{-1}}$ |
| $p$, $p_0$ | pressure, pressure at ground level | Pa |

| | | |
|---|---|---|
| $P$ | chemical production term | $\mathrm{mol\,m^{-3}\,s^{-1}}$ |
| $q$ | specific humidity | $\mathrm{kg\,kg^{-1}}$ |
| $r$ | total (i.e. humidified) particle radius | m |
| $\dot{r}$ | particle growth | $\mathrm{m\,s^{-1}}$ |
| $r_{\mathrm{a}}$ | aerodynamic resistance | $\mathrm{s\,m^{-1}}$ |
| $r_{\mathrm{b}}$ | quasi-laminar resistance | $\mathrm{s\,m^{-1}}$ |
| $r_{\mathrm{c}}$ | surface resistance | $\mathrm{s\,m^{-1}}$ |
| $R$ | gas constant | $8.3144743\ \mathrm{J\,K^{-1}\,mol^{-1}}$ |
| $R_{\mathrm{a}}$ | specific gas constant for dry air | $287.048\ \mathrm{J\,K^{-1}\,kg^{-1}}$ |
| $R_{\mathrm{v}}$ | specific gas constant for water vapour | $461.523\ \mathrm{J\,K^{-1}\,kg^{-1}}$ |
| $S_{\mathrm{h}}, S_{\mathrm{m}}$ | stability functions for buoyancy and shear production (see also $G_{\mathrm{h}}, G_{\mathrm{m}}$) | 1 |
| $S$ | chemical loss term, or sink | $\mathrm{s^{-1}}$ |
| $Sc$ | Schmidt number | 1 |
| $St$ | Stokes number | 1 |
| $S_{\mathrm{r}}$ | supersaturation at the droplet's surface | 1 |
| $S_{\infty}$ | ambient supersaturation | 1 |
| $t$ | time | s |
| $T$ | temperature | K |
| $u, v$ | west–east and north–south horizontal wind component | $\mathrm{m\,s^{-1}}$ |
| $u_{\mathrm{g}}, v_{\mathrm{g}}$ | west–east and north–south horizontal geostrophic wind component | $\mathrm{m\,s^{-1}}$ |
| $u_*$ | friction velocity | $\mathrm{m\,s^{-1}}$ |
| $\bar{v}$ | mean molecular speed | $\mathrm{m\,s^{-1}}$ |
| $v_{\mathrm{g}}^{\mathrm{dry}}$ | dry deposition velocity of gases | $\mathrm{m\,s^{-1}}$ |
| $v_{\mathrm{a},i}^{\mathrm{dry}}$ | dry deposition velocity of particles | $\mathrm{m\,s^{-1}}$ |
| $w$ | subsidence, i.e. vertical wind component | $\mathrm{m\,s^{-1}}$ |
| $w_{\mathrm{l},i}$ | dimensionless liquid water content ($V_{\mathrm{aq}}/V_{\mathrm{air}}$) of bin $i$ | 1 |
| $w_{\mathrm{t}}$ | sedimentation or terminal velocity | $\mathrm{m\,s^{-1}}$ |
| $z$ | vertical coordinate, positive upwards | m |
| $z_0$ | roughness length | m |
| $\alpha$ | accommodation (or condensation) coefficient | 1 |
| $\alpha_{\mathrm{c}}$ | accommodation (or condensation) coefficient of water | 1 |
| $\Delta t$ | model time step | s |
| $\Theta$ | potential temperature | K |
| $\kappa$ | von Kármán constant | 0.4 |
| $\lambda$ | mean free path length | m |

| $\nu$ | kinematic viscosity of air | $\mathrm{m^2\,s^{-1}}$ |
| $\rho$ | air density | $\mathrm{kg\,m^{-3}}$ |
| $\rho_\mathrm{s}$ | saturation water vapour density | $\mathrm{kg\,m^{-3}}$ |
| $\rho_\mathrm{w}$ | water density | $\mathrm{1000\,kg\,m^{-3}}$ |
| $\sigma_\mathrm{X}$ | absorption cross section of species X | $\mathrm{m^2}$ |
| $\phi_\mathrm{X}$ | quantum yield of species X | 1 |
| $\Phi_\mathrm{s}$ | stability function for aerodynamic resistance calculation | 1 |

## Appendix B: List of abbreviations

| DMS | dimethylsulfide ($\mathrm{CH_3{-}S{-}CH_3}$) |
| ESM | Earth System Model |
| EUPL | European Union Public Licence |
| FT | free troposphere |
| KPP | the Kinetic PreProcessor |
| LWC | liquid water content |
| MBL | marine boundary layer |
| MIFOG | MIcrophysical FOG model |
| netCDF | network Common Data Form |
| PIFM | Practical Improved Flux Method (radiative code) |
| TKE | turbulence kinetic energy |

## Appendix C: Subsidence profiles

Three subsidence profiles are currently implemented in MISTRA-v9.0.

- – Option 1 follows the hyperbolic expression of Bott et al. (1996, Eq. (5)).

- – Option 2 linearly decreases from `wmin` at ground level to `wmax` at height = `etaw1`.

- – Option 3 linearly decreases from `wmin` at ground level to `wmax` at height = `zinv`.

*Author contributions.* From the early 2000's to 2015, development and application of the branch of MISTRA presented in this paper took place under the lead of Roland von Glasow (RvG). JB updated the model code and performed model runs. MT performed additional model

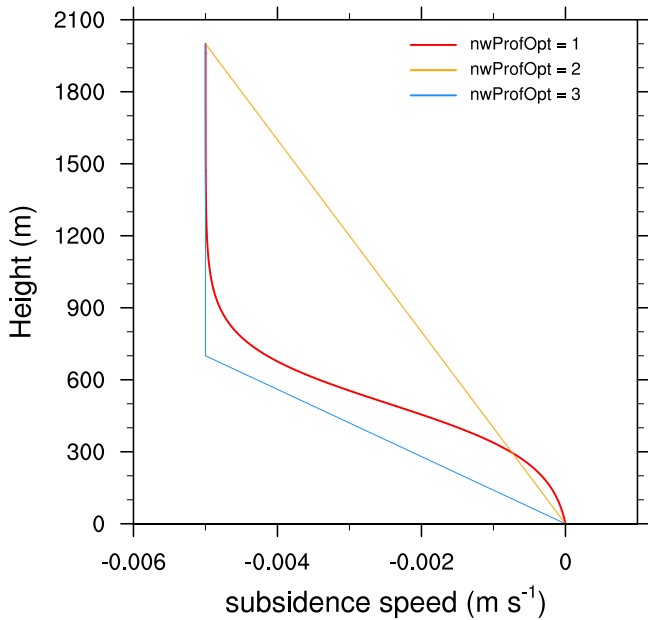

**Figure C1.** Vertical profiles of subsidence, computed for $w_{\min} = 0\ \mathrm{cm\,s^{-1}}$, $w_{\max} = -0.5\mathrm{cm\,s^{-1}}$, and $z_{\mathrm{inv}} = 700$ m.

runs. JK designed and led the project. AB provided help with the model. JB wrote the first draft of the paper. All co-authors commented and provided significant inputs to subsequent revisions. The technical description of the model (Sect. 2) was mostly written by RvG and adapted
for the current publication. This paper is dedicated to the memory of Roland von Glasow who sadly died before the final version could be prepared.

*Competing interests.*  The authors declare that they have no conflict of interest.

*Disclaimer.*  Unless required by applicable law or agreed to in writing, software distributed under the EUPL-v1.1 Licence is distributed on an "as is" basis, without warranties or conditions of any kind, either express or implied.

*Acknowledgements.*  We would like to thank Rolf Sander for his help with the update of KPP. We thank people from UEA, especially Claire Reeves, who got involved and provided help with regards the ASIBIA project after Roland's passing. We acknowledge all people who contributed to the development of previous versions of MISTRA and/or used the model in their work: A. Aiuppa, A. G. Allen, S. Arellano, S. Bleicher, N. Bobrowski, P. Bräuer, N. Brought, J. B. Burkholder, J. Buxmann, Z. Buys, L. J. Carpenter, P. J. Crutzen, W. D'Alessandro, J. E. Dibb, R. M. Dunk, B. D. Finley, A. Franco, B. Galle, G. B. Giuffrida, A. Held, K. E. Hornsby, L. G. Huey, O. W. Ibrahim, S. Inguaggiato,
485   M. Johansson, A. E. Jones, C. E. Jones, P. L. Joyce, W. C. Keene, A. Kerkweg, J. Landgraf, M. J. Lawler, B. Lefer, J. Liao, I. Louban, P. E.

Loughlin, E. R. Lovejoy, B. Luo, T. A. Mather, G. McFiggans, J. Ofner, W.-G. Panhans, S. Pechtl, M. Piot, U. Platt, A. A. P. Pszenny, D. M. Pyle, K. A. Read, E. S. Saltzman, R. Sander, G. Schmitz, W. Schneider, U. Sievers, W. R. Simpson, L. Smoydzin, R. Sommariva, J. Stutz, D. J. Tanner, D. Tedesco, J. L. Thomas, K. Toyota, T. Trautmann, M. Valenza, T. Winterrath, M. Yalire, W. Zdunkowski, C. Zetzsch. Until 2015, the development of MISTRA was financially supported by several grants acknowledged in the papers cited in Sect. 1.2.

JB is grateful to Météo France and CNRS-EDYTEM for granting time to finalise this work.

We thank Rolf Sander and an anonymous reviewer for their constructive remarks that helped improving this paper, and Linda Smoydzin, Claire Reeves, Roberto Sommariva, and Peter Bräuer for their comments. We are thankful to Holger Tost who edited our manuscript.

*Financial support.* This research has been supported by the European Research Council under the European Union's Seventh Framework Programme (FP7-2007-2013, ASIBIA project, grant agreement no. 616938), and the Horizon 2020 Research Infrastructure EUROCHAMP-
2020 (grant no. 730997).

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
