# Peer review of "A description of the first open source community release of MISTRA-v9.0: a 0D/1D atmospheric boundary layer chemistry model"

_Geoscientific Model Development, 2021_

## Referee Comment (RC1)

**Review by Rolf Sander**

Bock et al. present the 1D tropospheric chemistry model MISTRA-v9.0 as an open-source community release. I strongly support that this previously closed source code is now made available to the research community. I recommend publication of the manuscript in GMD after considering several comments as described below.

**Specific comments**

- Title: In the title, MISTRA is called an "atmospheric model". However, as there is apparently no code for the stratosphere or the upper atmosphere, it may be better to call MISTRA a "tropospheric model".

- Section 1.1: The advantages of a 1D model compared to a 3D model are described in detail. Maybe a short comparison of MISTRA to 0D (box) models could be added as well.

- p. 2, l. 55: I suggest to change "halogen chemistry" to "tropospheric halogen chemistry". Otherwise, the reader might expect stratospheric ozone hole chemistry as well.

- Figure 1: Only DMS emissions are shown here but iodine species can also be emitted in the model.

- Section 2.3, ll. 168-170: If, outside of clouds, the term "aqueous phase" is used only for sub-cloud aerosol, does this mean that there is no aerosol above the clouds?

- Section 2.3.1, Equation (10): This is the central and most important part of the chemistry code for the gas phase. As such, I think it deserves to be described in more detail. All terms should be explained in the order in which they appear in the equation.

- Section 2.3.1, Equation (10): The chemical loss of a species is proportional to its concentration, therefore the loss term includes $c_g$ as a factor. However, why is the deposition $D$ not multiplied by the concentration $c_g$?

- Section 2.3.1: When I calculate the "mean transfer coefficient" for a monodisperse aerosol using equation (11), I get a different value than with equation (12). This is because equation (12) includes the liquid water content and equation (11) does not. Thus, the equations produce different quantities, and $\overline{k_t}$ should not be called the "mean" of $k_t$.

- Section 2.3.1: "The last term in equation (10) describes the transport from the gas phase into the aqueous phases [...]"

  It describes not only the transport *into* the aqueous phases but also the reverse process, i.e., *out of* the aqueous phases.

- Section 2.3.2: When the unit mol/m$^3$ is used for aqueous-phase concentrations, it would be important to mention if it refers to 1 m$^3$ of air or to 1 m$^3$ of solution.

- Section 2.3.6: It should be explained how the deposition $D$ in equation (10) is calculated from the dry deposition velocity.

- As I have been directly mentioned in the Community comment by Roberto Sommariva, I would like to add my view as well: I agree that co-authorship should be offered to all model developers who contributed code which is now converted to open source. However, I think it is necessary to distinguish between model users and model developers. Contributions of other colleagues need to be checked individually and co-authorship should be offered where applicable. As far as I know, Roberto Sommariva and Susanne Pechtl have made substantial code contributions (mechanism update and iodine chemistry, respectively). Roberto Sommariva also mentions my contributions: The first halogen mechanism in MISTRA was taken from Sander and Crutzen (1996), and the KPP code has been presented by Sandu and Sander (2006). It is sufficient for me if these two papers are cited. I do not claim authorship for the current manuscript.

**Technical Comments**

- Section 1.2: The acronyms MBL and PIFM should be explained when they are used for the first time.

- p. 3, l. 79: When KPP is introduced, I suggest to cite the KPP model description by Sandu & Sander (2006, doi:10.5194/ACP-6-187-2006). Note that I have to declare a COI here because I'm a co-author of that paper.

- p. 6, ll. 138-139: Something is wrong with the sentence "after Davies (1985) Bott et al. (see also 1996)".

- Section 2.3.1, Equation (10): Why is the symbol $S$ used for the loss term? I suggest to use the same symbol as in equation (13), i.e., the symbol $L$.

- Section 2.3.1: Both $K_h^{cc}$ and $H_s^{cc}$ are used for the dimensionless Henry constant. I suggest to use only the symbol $H_s^{cc}$. In the context of equation (10), this also avoids confusion with $K_h$, the turbulent exchange coefficient for heat.

- Section 2.3.5 and Appendix A: $J$ is the photolysis rate constant, not the photolysis rate.

- Table 1: The term "netCDF" is mentioned in the table but not explained. I suggest to add a link to `https://www.unidata.ucar.edu/software/netcdf/` or `https://en.wikipedia.org/wiki/NetCDF`.

- Section 3.2.2: Ferret and NCL are mentioned here but not explained. I suggest to add a short explanation or a citation.

- Section 4.2: The term $NO_x$ should be defined.

- Appendix A: It is very good to have this list of symbols. It would be even more useful, if you can add the units that are used in MISTRA.

- Appendix A: For constants, their values could be shown as well ($R$, $R_a$, $R_v$, and maybe more)

- Appendix A: Several symbols should be added:

  - $g$
  - $H^*$ (effective Henry constant)
  - $H_s^{cp}$
  - $\overline{k_t}$
  - $M$ (molar mass)
  - St (Stokes number)

- Appendix B: Please add DMS, LWC, MIFOG and PIFM.

**References**

Sander, R. and Crutzen, P. J.: Model study indicating halogen activation and ozone destruction in polluted air masses transported to the sea, J. Geophys. Res., 101D, 9121–9138, doi:10.1029/95JD03793, 1996.

Sandu, A. and Sander, R.: Technical note: Simulating chemical systems in Fortran90 and Matlab with the Kinetic PreProcessor KPP-2.1, Atmos. Chem. Phys., 6, 187–195, doi:10.5194/ACP-6-187-2006, 2006.

---

## Community Comment (CC3)

In agreement with Roberto Sommariva, I have severe concerns regarding the publication of this manuscript under given circumstances.

License:
I agree with Roberto that a GPL license might be more appropriate. In addition, the authors of this manuscript want to make code publicly available which has however (at least partly) not been written by anyone of them.

Author contribution:
As the authors themselves emphasise that "the paper develops the branch of MISTRA based on von Glasow (2000)", it should be pointed out clearly and unmistakably (in the manuscript text itself as well as the author contribution section) that development and application of MISTRA took place under the lead of Roland von Glasow.

Overall, this manuscript appears to me to be half-hearty written. I explain in the following why I come to this conclusion:

Just listing a number of publications written based on simulations using MISTRA (p.2, l.54 - p.3 l.67): (i) is from my point of view not sufficient for publishing the code as the variety of scientific applications is only vaguely mentioned (ii) does not value the work of those people who worked intensely with MISTRA (in Roland von Glasow's working group, first at the university of Heidelberg, later at UEA in Norwich) and all of them contributed to the continuous development and improvement of MISTRA.

My own work is cited wrongly (p.3, l.63): One of my publications with Roland deals with organic surface coatings on sea salt aerosols (this is not mentioned at all) but in the manuscript both citations (Smoydzin and von Glasow, 2007 and 2009) refer to chemistry over the Dead Sea. In addition, I did not implement an "ocean model" into MISTRA; I wrote a code making it possible to calculate chemistry in a liquid medium (i.e. the Dead Sea) and to calculate air-sea exchange of gas phase species explicitly!

Calling Susanne Pechtl's work "an improvement of iodine chemistry" (p.7, l. 167) is - politely spoken - an understatement.
She developed a completely new aqueous phase iodine chemistry mechanism coupled to the gas phase (and the existing chemistry scheme) which was unique and new in atmospheric chemistry research - and which is an essential and outstanding part of MISTRA. In addition, she was the first who investigated the nucleation potential of iodine species and wrote this part of the code in MISTRA which the authors now want to make public.

Roberto Sommariva wrote a comment himself. He added a substantial part of code to MISTRA and must be offered co-authorship of the manuscript.

Matthias Piot significantly changed the MISTRA code to apply the model at Arctic conditions. This part of the model development deserves from my point of view more credit than just a citation without any further explanation of this work (neither a co-authorship).

I am sure, a careful review of the work done by Roland and his co-workers would reveal other code contributors.

A few comments following the review by Rolf Sander:

Rolf suggests to call MISTRA a "tropospheric model" instead of "atmospheric model". Though, this expression is even more wrong. MISTRA (the 1d column version) is a model of the atmospheric boundary layer. It cannot be used for studying chemistry in a column reaching from the surface into the free troposphere or even up to the tropopause as the upper boundary conditions are not suitable for such an application (from a physical/numerical point of view).

This fact answers also one of his questions:"... does it mean that there are not aerosols above clouds?" If (boundary layer!) clouds are simulated, they usually reach to the top of the boundary layer, thus to the top/upper boundary of the column model.

Rolf further suggests to compare MISTRA with 0D box models: At first, it should be clearly pointed out, that MISTRA can be used both, as a box model (a comparison with CAABA/MECCA might be obvious) and a column model (which is rather unique).

As mentioned above, a more detailed description of the possible scientific applications of MISTRA (marine halogen chemistry in coastal regions, volcanoes, Arctic applications) would also be desirable as well a description of typical model setups (e.g. using the column or box in a pseudo-Lagrangian way as done in many studies discussed in the MISTRA publications).

A minor comment:
There is a typo in the reference of Joyce et. al (p.31, l.510)

---

## Author Comment (AC1)

We thank Rolf Sander for his positive evaluation of our manuscript, and numerous remarks that are very useful to improve the paper. His comments are reproduced below in black fonts, our answers are displayed in blue fonts.

**Specific comments**

- Title: In the title, MISTRA is called an "atmospheric model". However, as there is apparently no code for the stratosphere or the upper atmosphere, it may be better to call MISTRA a "tropospheric model".
  Thank you for your suggestion. However, we agree with Linda Smoydzin's comment that "tropospheric" would not be suited. Instead, we replaced "atmospheric" by "atmospheric boundary layer". The title now reads:
  *A description of the first open source community release of MISTRA-v9.0: a 0D/1D atmospheric boundary layer chemistry model*

- Section 1.1: The advantages of a 1D model compared to a 3D model are described in detail. Maybe a short comparison of MISTRA to 0D (box) models could be added as well.
  We added the following sentence in Section 1.1:
  *Ultimately, box models (0D) are designed to focus only on a single grid cell processes, further reducing the computing cost as compared to 1D models.*

- p. 2, l. 55: I suggest to change "halogen chemistry" to "tropospheric halogen chemistry". Otherwise, the reader might expect stratospheric ozone hole chemistry as well.
  We agree with this suggestion and modified the text accordingly.

- Figure 1: Only DMS emissions are shown here but iodine species can also be emitted in the model.
  Indeed MISTRA can account for any gas emission, DMS was just an important example of such emission. We agree that it was too reductive and maybe misleading. We thus replaced "DMS" in Figure 1 by "emissions (DMS, iodine, ...)".

- Section 2.3, ll. 168-170: If, outside of clouds, the term "aqueous phase" is used only for sub-cloud aerosol, does this mean that there is no aerosol above the clouds?
  In MISTRA, there are aerosols above the clouds, however the aqueous phase chemistry is computed only when a minimum liquid water content threshold is reached, which is never the case above the cloud top, or more generally above the top of the BL. To clarify, we added the following sentence after the one you cited:
  *Aqueous chemistry is not computed above the top of the boundary layer (i.e. the top of clouds, if present).*

- Section 2.3.1, Equation (10): This is the central and most important part of the chemistry code for the gas phase. As such, I think it deserves to be described in more detail. All terms should be explained in the order in

which they appear in the equation.

*We agree with your suggestion. We reorganised and expanded Section 2.3.1 so that each term of Equation (10) is described in the order it appears in the equation.*

- Section 2.3.1, Equation (10): The chemical loss of a species is proportional to its concentration, therefore the loss term includes cg as a factor. However, why is the deposition D not multiplied by the concentration $c_g$?

  *The dry deposition is indeed computed with a multiplication by the gas concentration (routine sedc in MISTRA). This was implicitly included in the deposition term "D", but we agree this writing was not consistent with the way it was written for the chemical loss. We thus explicitly reformulated the dry deposition term as $Dc_g$ in Equation (10).*

- Section 2.3.1: When I calculate the "mean transfer coefficient" for a monodisperse aerosol using equation (11), I get a different value than with equation (12). This is because equation (12) includes the liquid water content and equation (11) does not. Thus, the equations produce different quantities, and kt should not be called the "mean" of kt.

  *We agree with you. We thus rephrased the sentence presenting Equation (12) as follows:*

  *The transfer coefficient $\overline{k_t}$ for a particle population is given by the integral: [...].*

- Section 2.3.1: "The last term in equation (10) describes the transport from the gas phase into the aqueous phases [. . . ]"

  It describes not only the transport *into* the aqueous phases but also the reverse process, i.e., *out of* the aqueous phases.

  *We agree the wording was not correct. We rephrased this sentence that now reads:*

  *The last term in equation (10) describes the transport between the gas phase and the aqueous phases [...]*

- Section 2.3.2: When the unit mol/m³ is used for aqueous-phase concentrations, it would be important to mention if it refers to 1 m³ of air or to 1 m³ of solution.

  *We agree this is an important specification. The unit of aqueous-phase concentration is now clarified, the sentence reads:*

  *In each of these bins the following prognostic equation is solved for each chemical species $c_{a,i}$ (amount per air volume), [...].*

- Section 2.3.6: It should be explained how the deposition D in equation (10) is calculated from the dry deposition velocity.

  *We added the following at the end of Section 2.3.6:*

  *Finally, the dry deposition term D is calculated as:*

$$D = \exp\left(-\Delta t/h \times v_g^{\mathrm{dry}}\right) \tag{23}$$

  *where $\Delta t$ is the model time step, and $h$ is the height of the lowermost*

*model layer.*

- As I have been directly mentioned in the Community comment by Roberto Sommariva, I would like to add my view as well: I agree that co-authorship should be offered to all model developers who contributed code which is now converted to open source. However, I think it is necessary to distinguish between model users and model developers. Contributions of other colleagues need to be checked individually and co-authorship should be offered where applicable. As far as I know, Roberto Sommariva and Susanne Pechtl have made substantial code contributions (mechanism update and iodine chemistry, respectively). Roberto Sommariva also mentions my contributions: The first halogen mechanism in MISTRA was taken from Sander and Crutzen (1996), and the KPP code has been presented by Sandu and Sander (2006). It is sufficient for me if these two papers are cited. I do not claim authorship for the current manuscript.

  Thank you for clarifying your point of view regarding your own involvement in MISTRA.

  During the review process, we contacted several people who contributed to important parts of code in MISTRA, as detailed in our reply to Roberto Sommariva. Those who replied do not claim authorship for this paper and agree with the EUPL licensing.

**Technical Comments**

- Section 1.2: The acronyms MBL and PIFM should be explained when they are used for the first time.

  MBL was already explained at line 43.

  We added the meaning of PIFM acronym, plus the reference of the paper (already cited elsewhere in our manuscript) discussing this radiative code (Zdunkowski et al., 1982). The sentence starting at line 47 now reads:

  *The radiation code used in MISTRA, called PIFM1 (Practical Improved Flux Method, developed by Zdunkowski et al., 1982), was updated by Loughlin et al. (1997) and the new radiation code, PIFM2, was evaluated.*

- p. 3, l. 79: When KPP is introduced, I suggest to cite the KPP model description by Sandu & Sander (2006, doi:10.5194/ACP-6-187-2006). Note that I have to declare a COI here because I'm a co-author of that paper. Thank you, this reference was indeed missing. We added it as suggested, as long as the first paper presenting KPP (Damian et al., 2002). The sentence p. 3, l. 79 now reads:

  *The chemical "Kinetic PreProcessor" (KPP: Damian et al., 2002; Sandu and Sander, 2006) has been updated to the latest version 2.2.3 (`https://people.cs.vt.edu/~asandu/Software/Kpp/` last accessed 23 June 2021).*

- p. 6, ll. 138-139: Something is wrong with the sentence "after Davies (1985) Bott et al. (see also 1996)".

*Thank you for noticing this, we corrected the citation command. The end of the sentence now reads:*
*after Davies (1985) (see also Bott et al., 1996).*

- Section 2.3.1, Equation (10): Why is the symbol S used for the loss term? I suggest to use the same symbol as in equation (13), i.e., the symbol L.
  *There was indeed an inconsistency, thank you for your careful reading. We now choose to use the symbol $S$ (as "sink") in both equations (10) and (13) since $L$ is used elsewhere for the latent heat of condensation.*

- Section 2.3.1: Both $K_{\mathrm{h}}^{cc}$ and $H_{\mathrm{s}}^{cc}$ are used for the dimensionless Henry constant. I suggest to use only the symbol $H_{\mathrm{s}}^{cc}$. In the context of equation (10), this also avoids confusion with $K_{\mathrm{h}}$, the turbulent exchange coefficient for heat.
  *We agree and modified the symbol accordingly in equation (10).*

- Section 2.3.5 and Appendix A: J is the photolysis rate constant, not the photolysis rate.
  *Thank you, we corrected this in Section 2.3.5 and Appendix A.*

- Table 1: The term "netCDF" is mentioned in the table but not explained. I suggest to add a link to https://www.unidata.ucar.edu/software/netcdf/ or https://en.wikipedia.org/wiki/NetCDF.
  *We followed your suggestion, and explained the netCDF acronym in Table 1. The added footnote in this Table reads:*
  *[a] network Common Data Form, see https://www.unidata.ucar.edu/software/netcdf/ (last accessed 21/11/2021).*

- Section 3.2.2: Ferret and NCL are mentioned here but not explained. I suggest to add a short explanation or a citation.
  *We added the reference for NCL and the link to Ferret webpage. The sentence now reads:*
  *Plotting scripts provided as example are written for Ferret (`http://ferret.pmel.noaa.gov/Ferret/`, last accessed 04/11/2021)and NCL (NCAR, 2019), but neither are necessary to run the model.*

- Section 4.2: The term $NO_x$ should be defined.
  *The sentence now reads:*
  *An emission scenario of $NO_x$ ($NO+NO_2$) was defined (...).*

- Appendix A: It is very good to have this list of symbols. It would be even more useful, if you can add the units that are used in MISTRA.
  *Thank you for your feedback, we added the units in Appendix A.*

- Appendix A: For constants, their values could be shown as well (R, Ra, Rv, and maybe more)
  *We shown the numerical values for these constants, and for $c_p$, $g$, $\kappa$, and $\rho_{\mathrm{w}}$ in Appendix A.*

- Appendix A: Several symbols should be added:

- $g$
- $H^*$ (effective Henry constant)
- $H_s^{cp}$
- $\overline{k_t}$
- M (molar mass)
- St (Stokes number)

Thank you for your careful review, we added these symbols in Appendix A

- Appendix B: Please add DMS, LWC, MIFOG and PIFM.
  We have added these four abbreviations in Appendix B.

**References**

Damian, V., Sandu, A., Damian, M., Potra, F., and Carmichael, G. R.: The kinetic preprocessor KPP-a software environment for solving chemical kinetics, Computers & Chemical Engineering, 26, 1567–1579, https://doi.org/10.1016/S0098-1354(02)00128-X, 2002.

Zdunkowski, W. G., Panhans, W.-G., Welch, R. M., and Korb, G.: A radiation scheme for circulation and climate models, Contributions to Atmospheric Physics, 55, 215–238, 1982.

---

## Author Comment (AC2)

We thank Reviewer #2 for their positive evaluation of our manuscript, and numerous remarks that are very useful to improve the paper. R2's comments are reproduced below in black fonts, our answers are displayed in blue fonts.

**Remarks**

Title: I find the term "chemistry" in the title a bit reductive, as the model can also represent some aerosol processes, like particle nucleation. You might consider extending the title.

Thank you for your suggestion to extend the paper's title to also mention the aerosol processing aspects. However, the title is already long, all the more so since we extended it following the comments by Rolf Sander and Linda Smoydzin, to clarify that MISTRA is both 0D and 1D, and that it focuses on the boundary layer. The title now reads:

*A description of the first open source community release of MISTRA-v9.0: a 0D/1D atmospheric boundary layer chemistry model*

L13: you might cite also Bellouin et al. (Rev. Geophys. 2020, `doi:10.1029/2019rg000660`), for a more recent assessment.

We added this reference, the sentence now reads:

*They significantly affect the radiative balance of the atmosphere, through direct (scattering and absorption) and indirect effects (cloud properties modification) (Carslaw et al., 2010; Boucher et al., 2014; Bellouin et al., 2020).*

L17: please add some references for these two other effects you mention.

We added references for these two effects, the sentence now reads:

*Other impacts include the reduction of visibility (see for instance Seinfeld and Pandis (2016, Chap. 15); Zhang et al. (2020) and ref. therein) and health effects of pollution (e.g. Pöschl (2005), Molina et al. (2020) and ref. therein).*

L46: *In this work*, do you mean Bott (1997)? I would be more specific, since "this work" could mean the present manuscript.

"this work" indeed referred to the study of Bott (1997). We merged both sentences to clarify, it now reads:

*Based on this first version of MISTRA, Bott (1997) further included typical particle distributions of urban and rural aerosols for the study of MBLs influenced by continental air masses, and assessed the radiative forcing of stratiform cloud.*

L76: can you provide some example of the strict coding rules mentioned here?

The most prevalent change in the model code is the explicit declaration of all variables, to replace `implicit double precision (a-h,o-z)` that was used so far. Other examples include the rewriting of obsolete features such as arithmetic if, and go to statements, which are strongly discouraged in modern code.

We believe that these technical examples do not need to be provided in the manuscript. Instead, we added two references to practical guidelines for Fortran coding, which include the aforementioned examples. Though these guidelines are only published online, they are often referred to in the atmospheric sciences modelling community, and we think they deserve to be cited in the manuscript. The sentence now reads:

*To improve robustness and portability of the code, intensive controls throughout the code have been performed to track issues, fix bugs, and conform to strict coding rules (Metcalf et al., 2004) and coding standards (see for instance $http:$ $//www.\,umr-cnrm.\,fr/\,gmapdoc/\,IMG/\,pdf/\,coding-rules.\,pdf$ and $http:\,//$ $www.\,reading.\,ac.\,uk/physicsnet/units/\,3/\,3phss/\,F90Style.\,pdf$, last accessed 26/10/2021)*

L77: how would you ensure future maintainability if the Forcheck tool is no longer distributed?
Forcheck is no longer distributed, but any current user with a valid licence can still use it. Furthermore, using such a tool (other might exist as well, and regular Fortran compilers also perform lexical analysis) is not mandatory to maintain the model code.

L79: please provide a reference to KPP (I think this is Sandu and Sanders, 2006, doi:10.5194/acp-6-187-2006).
The reference to KPP papers was indeed missing, thank you for noticing this. We added the reference you suggested, plus the paper describing the first version that was used in MISTRA before we updated it. In the same sentence, we also clarified that a version of KPP specifically tuned for MISTRA is provided along with the model code. The sentence now reads:
*The chemical "Kinetic PreProcessor" (KPP: Damian et al., 2002; Sandu and Sander, 2006) has been updated to the latest version 2.2.3 ($https:\,//people.$ $cs.\,vt.\,edu/\,\sim asandu/Software/Kpp/$ last accessed 23 June 2021) with minor tuning for use in MISTRA (see the* Code availability *section at the end of the paper).*

L92: all model layers: how many? Is this configurable? Please clarify.
We added the following text in Sect. 2.1:
*The vertical grid is separated into three regions: the lowest part is made of 100 layers with a constant thickness of 10 m, followed by 50 layers with logarithmically equidistant layers up to 2000 m height. The third region is a constant atmosphere whose characteristics are based on the standard atmosphere. It extends up to 50 km height and is only used for radiation calculations. These vertical grid settings (number and thickness of layers) can be easily configured as required.*

L94: *Fluxes of seasalt... are included*, I would add "(see Sect. 2.3.6)".
We added this reference to Sect. 2.3.6.

L95: could you elaborate a bit more on the nucleation module? How is this process parametrized?
We added a subsection 2.3.7 to present the nucleation module, with the following description:
*A module computing the nucleation process was implemented in MISTRA by Pechtl et al. (2006). Only a brief overview is given here, while a comprehensive description is given in the model manual (Chapter 4). The nucleation module developed by Pechtl et al. (2006) includes both ternary sulfuric acid-ammonia-*

*water ($H_2SO_4$ – $NH_3$ – $H_2O$) nucleation, and homomolecular homogeneous OIO nucleation. The former is explicitly calculated as a function of $H_2SO_4$ and $NH_3$ concentrations, relative humidity, and temperature following the work by Napari et al. (2002). The latter is parameterised following Burkholder et al. (2004). Each process can be activated or not independently (see Table 1), and lead to the computation of "real" nucleation rates. In a second step, the "apparent" nucleation rate is computed after the work of Kerminen and Kulmala (2002) and Kerminen et al. (2004).*

*The nucleated particles computed in this module can then be integrated in the model, with three possible options: (i) no coupling, (ii) coupling with the microphysics without feedback on chemistry, and (iii) coupling with microphysics and chemistry (see Table 1).*

L126-127: What about nucleation? Newly nucleated particles can have size below 5 nm, hence outside this range.
*Indeed, the thermodynamic stable clusters are about 1 nm in diameter. This is why an "apparent" nucleation rate of larger particles is computed from the "real" nucleation rate by means of an analytical formula.*

L295-296: *Note that default values are for all of them, however they should be systematically redefined by the user to match the simulated atmosphere.* I am not sure I understand this sentence, could you be more explicit?
*We rephrased this sentence, that now reads:*
*All these parameters have default values, even if most of them are expected to be redefined by the user to match the simulated atmosphere.*

L297: still, it would be interesting to know the temporal coverage of a typical run.
*We added the following sentence:*
*Typical run duration covers a few hours to a few days. Longer run duration is sometimes necessary for model spin up. The restart option of the model allows a single spin up run to initialise the model, and perform a sensitivity analysis from that stage, for instance.*

Sect. 3.1: I would not use subsections here, they are too short anyway.
*The text within sections 3.1.x is indeed short, but tables are large and include significant piece of information. We kept the current headers for the sake of clarity of the manuscript, but now used un-numbered subsections to lighten them.*

Conclusions: this is quite short. You could extend it, for example, by summarizing again the main capabilities/scope of the model and by adding a few sentences about current plans for model extension/improvement.
*We extended the conclusion with the following description:*
*MISTRA-v9.0 is a versatile model with a range of capabilities, from the study of status cloud microphysics, radiative forcing and turbulence, to the mutiphase atmospheric chemistry of the boundary layer. While its original purpose was*

*only the study of cloud-free and cloudy marine boundary layer, MISTRA was successfully extended in previous studies to model other environments such as polar conditions and volcanic plumes. In this study, we updated the model code to comply with coding standards, [...].*

**Corrections**

L15: large area –> large surface area.
Corrected

L23: limited area –> limited domain.
Limited-Area Models (LAMs) is the common wording, see for instance de Elía et al. (2002); Davies (2014) or `https://www.ecmwf.int/en/about/media-centre/news/2017/experts-debate-progress-limited-area-modelling` (last accessed 10/10/2021).

L28: *physic* –> *physics.*
Corrected

L28: *is* –> *are.*
Corrected

L60: *box mode* –> *box model.*
It now reads *box-model mode*

L124: *water is present* –> *water were present.*
Corrected

L125: *minimum aerosol radius* –> *minimum aerosol dry radius* (I guess).
Changed to *minimum dry aerosol radius.*

L137: better "time integration"?
This is indeed better, thank you.

L138: I think you mean "see also Bott (1996)".
We corrected the parenthesis for this reference.

L163: it is actually "on aerosol" and "in cloud particles".
In this context, "aerosol" refers to deliquescent aerosol particles, inside which bulk chemical reactions are accounted for. Surface reactions occurring on aerosol particles are also accounted for in the MISTRA model, as described in the following sentence. We thus rephrased the sentence L163 that now reads:
*The multiphase chemistry module comprises chemical reactions in the gas phase as well as in deliquescent aerosol and cloud particles.*

L166: DMS acronym not defined.
It now reads *"(...) of the oxidation of dimethylsulfide (DMS)."*, and we added DMS in Appendix B.

L203: I would use the term "coagulation" instead of "collisions".
We used coagulation as you suggested.

L315: *mandatory –> required.*
Changed

L319: please add the references or the links for ferret and NCL.
We added the reference for NCL and the link to Ferret webpage. The sentence now reads:
*Plotting scripts provided as example are written for Ferret (`http: // ferret. pmel. noaa. gov/ Ferret/`, last accessed 04/11/2021)and NCL (NCAR, 2019), but neither are necessary to run the model.*

L385: please append "(Fig. 6a)" at the end of the sentence.
We added this internal reference at the end of the sentence.

Figure 2 caption: as function –> as a function.
Corrected

Figure 3: please use the same contour levels for top and bottom panel (as you do in Fig. 4, for example).
We now increased the maximum contour level to 0.7 instead of 0.6. We added a sentence to clarify the apparent difference regarding the minimum contour level value "*Note the minimum contour level is set to 0.01 in both panels, but was displayed incorrectly in the original figure.*"

Figure 6 caption: please add that the MISTRA-v9.0 is also "without collision-coalescence".
the text now reads "*MISTRA-v9.0 (without collision-coalescence implemented)*" to make clear that this is not an option currently available in MISTRA-v9.0 (even is this was already highlighted in the text).

Figure 8: Scales are identical for both, actually the top right scale goes to 60 instead of 59. Not a big difference, but I would fix it.
Fixed

Eq. (12): the "lg" notation for the logarithm could be ambiguous, please specify the base or use "ln" if natural log.
"lg" is the notation for the decadic logarithm $log_{10}$, as recommended by the IU-PAC Green Book (iupac.org/greenbook) and the SI Brochure (`www.bipm.org/ en/publications/si-brochure`), both of which GMD authors are asked to follow (`www.geoscientific-model-development.net/submission.html#math`).

**References**

Davies, T.: Lateral boundary conditions for limited area models, Quarterly Journal of the Royal Meteorological Society, 140, 185–196, https://doi.org/ 10.1002/qj.2127, 2014.

de Elía, R., Laprise, R., and Denis, B.: Forecasting skill limits of nested, limited-area models: a perfect-model approach, Monthly Weather Review, 130, 2006–

2023, https://doi.org/10.1175/1520-0493(2002)130⟨2006:FSLONL⟩2.0.CO;2, place: Boston MA, USA Publisher: American Meteorological Society, 2002.

---

## Author Comment (AC3)

We thank Linda Smoydzin for her thorough reading, and comments on the manuscript. Her comments are reproduced below in black fonts, our answers are displayed in blue fonts.

License:
I agree with Roberto that a GPL license might be more appropriate. In addition, the authors of this manuscript want to make code publicly available which has however (at least partly) not been written by anyone of them.
GNU General Public License (GPL) and European Union Public Licence (EUPL) provide similar rules (authorisations and obligations) regarding the model code (see for instance `https://choosealicense.com/appendix/`). However, the EUPL provides wider compatibility with other licences, which means that MISTRA (released under EUPL) can be merged with code covered by a compatible license, such that the combined derivative work can be distributed under the compatible licence. We asked developers of legacy parts in the MISTRA-v9.0 code (A. Kerkweg: contribution to netCDF output format; B. Luo: ion activities; S. Pechtl: nucleation; J. Landgraf: photolysis rates) whether they agreed to release of the their contribution under the EUPL. The first three confirmed that they were happy with this. Unfortunately, despite repeated attempts via email and phone, we were unsuccessful in making contact with J. Landgraf.

Author contribution:
As the authors themselves emphasise that "the paper develops the branch of MISTRA based on von Glasow (2000)", it should be pointed out clearly and unmistakably (in the manuscript text itself as well as the author contribution section) that development and application of MISTRA took place under the lead of Roland von Glasow.
We modified the sentence cited above as follows:
*Our paper develops the branch of MISTRA of von Glasow (2000), whose development and application until 2015 took place under the lead of Roland von Glasow.* and we added the following sentence in the Author contribution section:
*From the early 2000s to 2015, development and application of the branch of MISTRA presented in this paper took place under the lead of Roland von Glasow.*

Overall, this manuscript appears to me to be half-hearty written. I explain in the following why I come to this conclusion:

Just listing a number of publications written based on simulations using MISTRA (p.2, l.54 - p.3 l.67): (i) is from my point of view not sufficient for publishing the code as the variety of scientific applications is only vaguely mentioned (ii) does not value the work of those people who worked intensely with MISTRA (in Roland von Glasow's working group, first at the university of Heidelberg, later at UEA in Norwich) and all of them contributed to the continuous development and improvement of MISTRA.
We understand the demand to present a thorough review of the capabilities of

MISTRA based on previous applications, and this is what we intended to do, not only by listing all publications based on MISTRA simulations, but also by providing detailed examples of use in a variety of situations (see section 4 of the paper). Since the principal objective of this paper is to present the updated version of MISTRA-v9.0, we believe this presentation is fully suited. Conversely, with over 25 papers published with MISTRA, a more extensive summary of the conclusions of each study would be beyond the scope of our work. Yet, we expanded the conclusion of the paper to give a summary of the model uses and potential.

My own work is cited wrongly (p.3, l.63): One of my publications with Roland deals with organic surface coatings on sea salt aerosols (this is not mentioned at all) but in the manuscript both citations (Smoydzin and von Glasow, 2007 and 2009) refer to chemistry over the Dead Sea. In addition, I did not implement an "ocean model" into MISTRA; I wrote a code making it possible to calculate chemistry in a liquid medium (i.e. the Dead Sea) and to calculate air-sea exchange of gas phase species explicitly!

Thanks you for pointing this out. We have revised the text in the manuscript. The sentences now reads:

*MISTRA was used to investigate the influence of organic coating at the surface of sea salt particles over boundary layer chemistry, and especially on bromine and chlorine chemistry in the aqueous phase (Smoydzin and von Glasow, 2007).*
and:
*MISTRA was also used to simulate the boundary layer chemistry over the Dead Sea after implementing a calculation of chemistry in this specific liquid medium and an explicit calculation of sea-air gas exchanges (Smoydzin and von Glasow, 2009).*

Calling Susanne Pechtl's work "an improvement of iodine chemistry" (p.7, l. 167) is - politely spoken - an understatement.

She developed a completely new aqueous phase iodine chemistry mechanism coupled to the gas phase (and the existing chemistry scheme) which was unique and new in atmospheric chemistry research - and which is an essential and outstanding part of MISTRA. In addition, she was the first who investigated the nucleation potential of iodine species and wrote this part of the code in MISTRA which the authors now want to make public.

We agree that the work by Susanne Pechtl is an outstanding part of MISTRA. Please note that her work is also described earlier in the manuscript as follows (p.2-3, l.55-56): "A major improvement was the introduction of a module for aerosol nucleation [...]".

We revised text to emphasise the importance and novelty of the work. It now reads: *A major development was the introduction of a module for aerosol nucleation which significantly improved the iodine chemistry (Pechtl et al., 2006, 2007).*

After the comment of R2, we also added a subsection (2.3.7) to give an overview of the nucleation module.

Regarding the open-source release of the nucleation code, we contacted Susanne

Pechtl and she agreed for this.

Roberto Sommariva wrote a comment himself. He added a substantial part of code to MISTRA and must be offered co-authorship of the manuscript.
As stated in the manuscript, Roberto Sommariva updated the gas phase mechanism in MISTRA. This update of the chemical mechanism was published in Sommariva and von Glasow (2012). This work is already public since the tables of reactions and rate constants are provided in the appendix of Sommariva and von Glasow (2012). We therefore feel citation is an appropriate recognition of that work. We contacted Roberto Sommariva to clarify if he wrote any unattributed part of the code but have to date received no response.

Matthias Piot significantly changed the MISTRA code to apply the model at Arctic conditions. This part of the model development deserves from my point of view more credit than just a citation without any further explanation of this work (neither a co-authorship).
The developments implemented by Matthias Piot in MISTRA are not part of MISTRA-v9.0. Similarly, Linda Smoydzin's developments related to the Dead Sea special case are not part of MISTRA-v9.0. Both their contributions were indeed kept separated in special versions of MISTRA (see the changelog in the model manual).

I am sure, a careful review of the work done by Roland and his co-workers would reveal other code contributors.
In our work, we paid great attention and significant efforts to document the code. We added headers in most (if not all) routines including the author name each time we could get this information. Following this review, we also added a CREDITS file in the Github repository, listing all co-authors of published papers.

A few comments following the review by Rolf Sander:
Rolf suggests to call MISTRA a "tropospheric model" instead of "atmospheric model". Though, this expression is even more wrong. MISTRA (the 1d column version) is a model of the atmospheric boundary layer. It cannot be used for studying chemistry in a column reaching from the surface into the free troposphere or even up to the tropopause as the upper boundary conditions are not suitable for such an application (from a physical/numerical point of view).
We agree that "tropospheric" would not be suited.
After R2 and Rolf Sander's comments, we revised the title to: *"A description of the first open source community release of MISTRA-v9.0: a 0D/1D atmospheric boundary layer chemistry model"*.

This fact answers also one of his questions:"... does it mean that there are not aerosols above clouds?" If (boundary layer!) clouds are simulated, they usually reach to the top of the boundary layer, thus to the top/upper boundary of the column model.

We agree with you, except the column model in MISTRA extends higher than the top of the simulated BL.

Rolf further suggests to compare MISTRA with 0D box models: At first, it should be clearly pointed out, that MISTRA can be used both, as a box model (a comparison with CAABA/MECCA might be obvious) and a column model (which is rather unique).
Actually this was already highlighted in several places:

- p.2 l.34-35: "MISTRA-v9.0 also includes a box-model configuration, which can be adapted for atmospheric simulation chamber applications."

- p.3 l.58-60: "[...] including alternative model configurations where the chemistry was computed in a zero-dimension (0D) atmospheric box mode"

- p.13 Table 1 (code switches to use the box-model mode)

- Section 4.3: comparison with a previous study using the box-model model

We further stressed this by rephrasing the first abstract sentence:
*We present MISTRA-v9.0, a one dimensional (1D) atmospheric chemistry model.*
which now reads:
*We present MISTRA-v9.0, a one dimensional (1D) and box (0D) atmospheric chemistry model.*

As mentioned above, a more detailed description of the possible scientific applications of MISTRA (marine halogen chemistry in coastal regions, volcanoes, Arctic applications) would also be desirable as well a description of typical model setups (e.g. using the column or box in a pseudo-Lagrangian way as done in many studies discussed in the MISTRA publications).
Following your comment and a comment by R2, we extended the Conclusion to summarise a few applications of MISTRA, developed in previous publications.

A minor comment:
There is a typo in the reference of Joyce et. al (p.31, l.510)
Thank you for your careful reading, we corrected the chemical expression which now displays "$NO_x$" instead of "\chem{NO_x}".

**References**

Sommariva, R. and von Glasow, R.: Multiphase halogen chemistry in the tropical Atlantic Ocean, Environmental Science & Technology, 46, 10 429–10 437, https://doi.org/10.1021/es300209f, 2012.

---

## Author Comment (AC4)

Answer to public comments by Roberto Sommariva, Claire Reeves and Peter Bräuer. We thank Roberto Sommariva, Claire Reeves, and Peter Bräuer for their comments, and for bringing several questions to our attention.

**Authorship and acknowledgements**

The first and main comment raised by Roberto Sommariva, and discussed in the other public comments, deals with the authorship about the model code, with the rather implicit claim that he should be offered co-authorship, along with other people who work with MISTRA in the past.
We can confirm that our manuscript conforms to the GMD authorship guidelines stated here:
`https://www.geoscientific-model-development.net/policies/publication_ethics.html`
*"All authors listed on a presented scientific work must have contributed a significant part to it. Vice versa, all persons who contributed to the presented work need to be named."*

With respect to the authorship of the code, we asked developers of legacy parts in the MISTRA-v9.0 code (A. Kerkweg: contribution to netCDF output format; B. Luo: ion activities; S. Pechtl: nucleation; J. Landgraf: photolysis rates) whether they agreed with the open-source release of the their contribution to MISTRA, under the EUPL. The first three confirmed that they were happy with this. Unfortunately, despite repeated attempts via email and phone, we were unsuccessful in making contact with J. Landgraf.

We had already added headers including the author names to the various routines in the code. We now also added a CREDITS file in the repository, to summarise all known people who developed and worked with MISTRA, based on the list of publications.

With regard to the work done by Roberto Sommariva in MISTRA: this work was published in 2012 (Sommariva and von Glasow, 2012) and the modified gas phase chemistry mechanism was made public in this paper. This work is acknowledged in our paper through the citation of Sommariva and von Glasow (2012).

**Financial support**

We now also acknowledge the ASIBIA grant, which supported the developments presented here. Other grants have only supported work with MISTRA presented elsewhere and have been acknowledged there.

**Licencing**

GNU General Public License (GPL) and European Union Public Licence (EUPL) provide similar rules (authorisations and obligations) regarding the model code (see for instance `https://choosealicense.com/appendix/`). However, the

EUPL provides wider compatibility with other licences, which means that MISTRA (released under EUPL) can be merged with code covered by a compatible license, such that the combined derivative work can be distributed under the compatible licence.

As mentioned above, the people we contacted agreed for the release of the code under EUPL. This release had also been advertised to all known people involved with MISTRA in 2017, when a beta version of the code of MISTRA-v9.0 was first released on GitHub. Roberto Sommariva and Peter Bräuer forked this code, which we have taken as an implicit endorsement and agreement.

**Other points raised by Peter Bräuer**

In his comment, Peter Bräuer add 3 more remarks about the current release of MISTRA code:

- The organisation name on GitHub, "MISTRA-UEA", was chosen more than 4 years ago because "MISTRA" was already taken. The repository itself is named only "MISTRA". The organisation could be renamed, but this would break the existing links to the GitHub repository.

- admin privileges on the GitHub repository are not necessary to use, contribute and collaborate to the project. To our knowledge, admin privileges are mostly limited to technical aspects of the repository: naming and deleting for instance.

- regarding the manual source: it will be shared to any developers upon request.

**References**

Sommariva, R. and von Glasow, R.: Multiphase halogen chemistry in the tropical Atlantic Ocean, Environmental Science & Technology, 46, 10 429–10 437, https://doi.org/10.1021/es300209f, 2012.

---

## Author Response (AR2)

Following the last discussion with the Editor, we addressed the remaining comments as follow:

- In the Abstract, we removed the following sentence:
  In the past 20 years, MISTRA has been used in over 25 studies to address a wide range of scientific questions.

- In the Author Contribution section, we added this sentence:
  This paper is dedicated to the memory of Roland von Glasow who sadly died before the final version could be prepared.

- In the Acknowledgements section, we added 2 sentences :
  We acknowledge all people who contributed to the development of previous versions of MISTRA and/or used the model in their work: A. Aiuppa, A. G. Allen, S. Arellano, S. Bleicher, N. Bobrowski, P. Bräuer, N. Brought, J. B. Burkholder, J. Buxmann, Z. Buys, L. J. Carpenter, P. J. Crutzen, W. D'Alessandro, J. E. Dibb, R. M. Dunk, B. D. Finley, A. Franco, B. Galle, G. B. Giuffrida, A. Held, K. E. Hornsby, L. G. Huey, O. W. Ibrahim, S. Inguaggiato, M. Johansson, A. E. Jones, C. E. Jones, P. L. Joyce, W. C. Keene, A. Kerkweg, J. Landgraf, M. J. Lawler, B. Lefer, J. Liao, I. Louban, P. E. Loughlin, E. R. Lovejoy, B. Luo, T. A. Mather, G. McFiggans, J. Ofner, W.-G. Panhans, S. Pechtl, M. Piot, U. Platt, A. A. P. Pszenny, D. M. Pyle, K. A. Read, E. S. Saltzman, R. Sander, G. Schmitz, W. Schneider, U. Sievers, W. R. Simpson, L. Smoydzin, R. Sommariva, J. Stutz, D. J. Tanner, D. Tedesco, J. L. Thomas, K. Toyota, T. Trautmann, M. Valenza, T. Winterrath, M. Yalire, W. Zdunkowski, C. Zetzsch. Until 2015, the development of MISTRA was financially supported by several grants acknowledged in the papers cited in Sect. 1.2.